# FedRKMGC: Towards High-Performance Gradient Correction-based Federated Learning via Relaxation and Fast KM Iteration

## Abstract

Federated learning (FL) enables multiple clients to collaboratively train machine learning models without sharing their local data, providing clear advantages in terms of privacy and scalability. However, existing FL algorithms often exhibit slow convergence, particularly under heterogeneous data distributions, resulting in high communication costs. To mitigate this, we propose FedRKMGC, a novel federated learning framework that integrates Gradient Correction with the classical Relaxation strategy and the fast Krasnosel'skiĭ–Mann (KM) acceleration method to enhance convergence. Specifically, the fast KM technique is applied during local training to speed up client updates, while a relaxation step is introduced during server aggregation to further accelerate global iterations. By integrating these complementary mechanisms, FedRKMGC effectively mitigates client drift and accelerates convergence, improving both training stability and communication efficiency. Extensive experiments on standard FL benchmarks demonstrate that FedRKMGC consistently achieves superior convergence performance and substantial communication savings compared to the existing state-of-the-art FL methods.

## 1 Introduction

Federated learning (FL) has emerged as a promising paradigm for distributed optimization, enabling multiple clients to collaboratively train machine learning models without centralizing their data (McMahan et al., 2017; Pfeiffer et al., 2023; Li et al., 2023; Liao et al., 2024; Mora et al., 2024; Chen et al., 2025). Although this framework provides clear advantages in terms of privacy and scalability, the convergence speed of existing SGD-based FL methods remains fundamentally limited. As a result, achieving satisfactory accuracy often requires a large number of communication rounds, substantially increasing communication overhead. This naturally raises the question: *can we leverage past training information to accelerate learning and improve the theoretical convergence rate?* To the best of our knowledge, research exploring this direction remains relatively limited.

Communication efficiency is widely recognized as a central concern in FL (Pfeiffer et al., 2023). Classical algorithms such as FedAvg (McMahan et al., 2017) reduce communication cost by allowing multiple local updates per round. Although effective in relatively simple scenarios, FedAvg often struggles under more realistic and challenging conditions, particularly when client data are non independent and identically distributed (non-IID), which is a key challenge in FL (Chen et al., 2025). In such heterogeneous settings, local models tend to drift away from the global trajectory (Li et al., 2020), resulting in unstable convergence or even divergence. This phenomenon, commonly referred as *client drift*, significantly slows training and complicates optimization. To address client drift, several methods have been proposed to stabilize local updates (Li et al., 2020; Karimireddy et al., 2020; Acar et al., 2021). For instance, FedProx (Li et al., 2020) adds a proximal term to penalize large deviations from the global model, SCAFFOLD (Karimireddy et al., 2020) employs control variates to reduce the gap between local and global gradients, and FedDyn (Acar et al., 2021) incorporates dynamic regularization to better align client objectives with the global trajectory. Although these methods are effective in mitigating drift, they primarily focus on stabilizing local updates, and their convergence speed remains constrained by the inherent upper bounds of SGD-type algorithms. In contrast, extensive studies in convex optimization have demonstrated that

effectively leveraging historical information can substantially improve the convergence of first-order methods (Krasnosel'skiǐ, 1955; Maulén et al., 2024; Lieder, 2021). Classical approaches such as the Krasnosel'skiǐ–Mann (KM) iteration (Krasnosel'skiǐ, 1955; Baillon & Bruck, 1996; Cominetti et al., 2014), relaxation acceleration (Hadjidimos, 1978; Eckstein & Bertsekas, 1992), and Halpern-type iterations (Halpern, 1967; Lieder, 2021) have long been used to speed up convergence in optimization and numerical analysis (Gaïtanos et al., 1983; Xiao et al., 2018; Sun et al., 2025; Park & Ryu, 2022b). More recently, advances such as the fast KM method (Boţ & Nguyen, 2023) have provided provable improvements for certain problems, accelerating the convergence rate from $O(1/\sqrt{T})$ to $O(1/T)$ with $T$ being the number of iterations. These strong convergence guarantees inspire us to explore how classical acceleration techniques from optimization can be adapted to design faster and more efficient federated learning algorithms, thereby alleviating the communication overhead inherent in FL.

Motivated by these insights, we introduce FedRKMGC, a federated learning framework that effectively leverages past training information, drawing inspiration from classical relaxation strategies and the fast Krasnosel'skiǐ–Mann acceleration technique. By jointly addressing the challenges of client drift and slow convergence, our approach provides a unified optimization perspective that enhances both stability and communication efficiency.

Our main contributions can be summarized as follows:

- *A unified framework for stability and acceleration.* We introduce FedRKMGC, a federated learning algorithm that integrates gradient correction with fixed point acceleration techniques, expanding beyond existing methods that focus mainly on drift mitigation.

- *A two-level acceleration mechanism.* FedRKMGC employs fast Krasnosel'skiǐ–Mann extrapolation at the client side to accelerate the local updates, and applies a global relaxation step at the server side to speed up the aggregated model update, jointly enhancing convergence efficiency and stability.

- *Extensive empirical validation.* We evaluate FedRKMGC on standard FL benchmarks, demonstrating faster convergence and significant communication savings compared with the state-of-the-art FL methods.

## 2 RELATED WORK

### 2.1 FEDERATED LEARNING

**FedAvg** (McMahan et al., 2017) is one of the earliest and most widely adopted algorithms in federated optimization. It performs multiple local stochastic gradient steps on each client and then aggregates the resulting models via simple averaging. Despite its simplicity and scalability, FedAvg suffers from the well-known problem of *client drift* under heterogeneous data, where local updates deviate from the global descent trajectory, leading to slow or even unstable convergence. This limitation has motivated a broad line of research on drift mitigation. A natural extension is **FedProx** (Li et al., 2020), which augments the local objective with a proximal term that penalizes large deviations from the global model. By softly constraining local optimization, FedProx improves stability under non-IID settings. However, it does not fully address inter-client variability. To better align local and global updates, **SCAFFOLD** (Karimireddy et al., 2020) introduces client-specific control variates that correct for gradient drift. Similarly, **FedDyn** (Acar et al., 2021) adds a dynamic regularizer for each client at each round to align global and local solutions. Building on these ideas, subsequent works have explored different strategies beyond simple proximal or variance-reduction approaches. For example, **FedADMM** (Zhou & Li, 2023) formulates federated learning as a constrained optimization problem and leverages the alternating direction method of multipliers (ADMM) to jointly handle local updates and global consistency. *However, most of these approaches focus on stabilizing the local updates without explicitly considering acceleration. In contrast, our proposed FedRKMGC integrates correction with fixed-point acceleration techniques, thereby addressing both stability and speed in a unified manner.*

## 2.2 Acceleration Techniques in Optimization

As model sizes and datasets continue to grow, improving the efficiency of first-order methods has attracted increasing attention in optimization. A natural perspective is to interpret iterative optimization procedures as fixed-point iterations of suitable operators. This perspective not only provides a clearer understanding of their convergence behavior but also enables the design of acceleration strategies at the operator level. Within this framework, one of the most widely studied schemes is the **Krasnosel'skiĭ–Mann (KM) iteration**,

$$x^{t+1} = (1 - s^t)x^t + s^t \mathcal{T}(x^t),$$

where $\mathcal{T}$ is a $\zeta$-averaged operator with $\zeta \in (0, 1]$ and $s^t \in (0, 1]$ (see, e.g., (Park & Ryu, 2022a) for the concept of such operators). The KM framework unifies many classical algorithms and provides a natural language for studying their asymptotic behavior. Classical analyses show that KM-type schemes typically converge sublinearly. For constant steps $s_t$, the fixed-point residual decreases at a rate of $o(1/\sqrt{T})$ (Baillon & Bruck, 1996); for more general nonconstant step sequences, upper bounds of $\mathcal{O}(1/\sqrt{T})$ were established in (Cominetti et al., 2014; Liang et al., 2016), with subsequent refinements obtaining $o(1/\sqrt{T})$ (Boţ & Csetnek, 2017; Bravo & Cominetti, 2018). Variants that incorporate inertial or momentum terms have been studied as well and largely inherit similar sublinear rates (Maulén et al., 2024). More recently, a line of work has proposed provably faster fixed-point accelerations. Notably, Boţ & Nguyen (2023) et al. designed a *fast KM* scheme of the form

$$\begin{aligned} x^{t+1} &= x^t + \frac{\gamma}{2(t+1+\gamma)}\big(\mathcal{T}(x^t) - x^t\big) + \frac{t+1}{t+1+\gamma}\big(\mathcal{T}(x^t) - \mathcal{T}(x^{t-1})\big) \\ &= \frac{2(t+1)+\gamma}{2(t+1+\gamma)}\big(x^t + \mathcal{T}(x^t)\big) - \frac{t+1}{t+1+\gamma}\mathcal{T}(x^{t-1}), \end{aligned}$$

where $\gamma \geq 2$. The fast KM can be interpreted as a KM update enhanced with an extrapolation term $\frac{t+1}{t+1+\gamma}\big(\mathcal{T}(x^t) - \mathcal{T}(x^{t-1})\big)$, which improves the theoretical convergence rate of the fixed-point residual to $o(1/T)$ under appropriate conditions. The strong convergence guarantee of the fast KM has made it attractive for a variety of applications. For example, it has recently been applied to optimization problems with linearly separable constraints (Sun et al., 2025), highlighting the potential of fixed-point acceleration as an efficient tool for improving optimization algorithms.

In parallel, **relaxation techniques** have long been employed as a simple yet effective means of enhancing convergence. By interpolating between the current iterate and the next update, relaxation can often yield substantial empirical speedups. Such techniques are widely used in areas ranging from solving linear systems to optimization problems (Eckstein & Bertsekas, 1992; Xiao et al., 2018; Sun et al., 2025).

*Motivated by these advances, we propose to integrate fast KM extrapolation at the client side with a global relaxation step at the server side, forming a two-level acceleration mechanism within federated learning. This combination brings together the strengths of fixed-point acceleration and relaxation, addressing both stability and efficiency in a unified framework.*

## 3 Proposed FedRKMGC

### 3.1 Overview

Many existing federated learning methods are based on stochastic gradient descent, whose inherent limitations for convergence rate often lead to high communication costs. This challenge is further exacerbated by heterogeneous data distributions across clients, which can slow down convergence and destabilize training. To accelerate model training and reduce the communication required to reach target accuracy, we propose an accelerated Gradient Correction-based federated learning framework (**FedRKMGC**) that leverages past training information, drawing inspiration from the fast **K**rasnosel'skiĭ–**M**ann iteration and **R**elaxation techniques.

Specifically, in each communication round, the server first broadcasts the current global model to the selected clients. Upon receiving it, each client performs local training (see ② in Figure 1). Instead of directly applying the raw gradients, it refines its update by subtracting a correction term (①) that captures the discrepancy between local and global trajectories from previous rounds. This correction evolves dynamically and is further accelerated using a fast KM iteration step (Boţ & Nguyen, 2023;

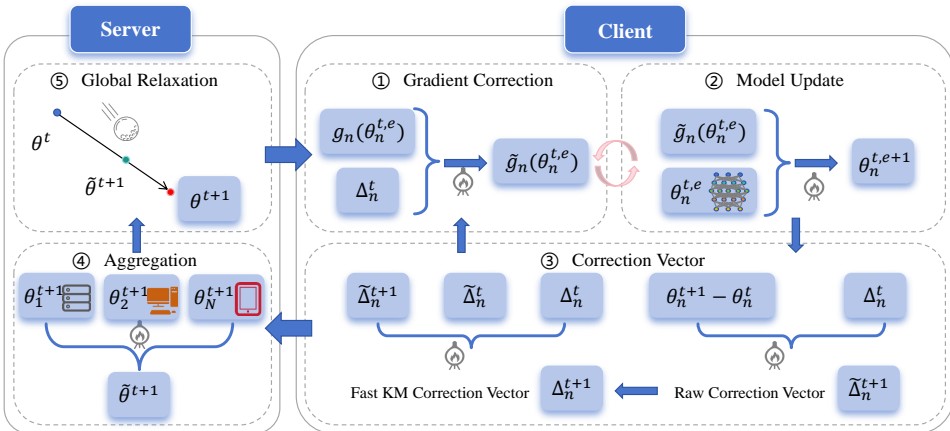

Figure 1: The framework for our proposed FedRKMGC.

Sun et al., 2025) (③), which incorporates historical gradient information to speed up alignment with the global direction. After local training, clients send their updated models back to the server. The server aggregates these updates as usual (④) and then applies a relaxation step that combines the new aggregate with the previous global iterate (⑤) to update the global model. This relaxation helps the global model advance more rapidly while maintaining stability.

By iteratively performing locally corrected updates and relaxed global aggregation, FedRKMGC effectively mitigates client drift and improves communication efficiency. This two-level acceleration mechanism, consisting of client-side corrections and server-side relaxation, will be described in detail in the following sections. The overall framework of FedRKMGC is shown in Figure 1, and the detailed algorithm is provided in Algorithm 1.

---

**Algorithm 1** FedRKMGC

---

1: **Initialization:** $\theta^0$, $\Delta_n^0 = \tilde{\Delta}_n^0 = \mathbf{0}$, $\forall n$
2: **for** $t = 0, 1, \ldots, T-1$ **do**
3:     Server broadcasts $\theta^t$ to sampled clients $\mathcal{C}_t$
4:     **for** each client $n \in \mathcal{C}_t$ **in parallel do**
5:         Set $\theta_n^{t,0} \leftarrow \theta^t$
6:         **for** $e = 0$ to $E-1$ **do**
7:             Corrected gradient: $\tilde{g}_n(\theta_n^{t,e}) \leftarrow g_n(\theta_n^{t,e}) - \Delta_n^t$
8:             Update: $\theta_n^{t,e+1} \leftarrow \theta_n^{t,e} - \eta\, \tilde{g}_n(\theta_n^{t,e})$
9:         **end for**
10:        $\theta_n^{t+1} \leftarrow \theta_n^{t,E-1}$
11:        Raw correction: $\tilde{\Delta}_n^{t+1} \leftarrow \Delta_n^t - \beta(\theta_n^{t+1} - \theta^t)$
12:        Fast KM correction: $\Delta_n^{t+1} \leftarrow \frac{2(t+1)+\gamma}{2(t+1+\gamma)}\big(\tilde{\Delta}_n^{t+1} + \Delta_n^t\big) - \frac{t+1}{t+1+\gamma}\tilde{\Delta}_n^t$
13:     **end for**
14:     Server aggregation: $\tilde{\theta}^{t+1} \leftarrow \frac{1}{|\mathcal{C}_t|}\sum_{n \in \mathcal{C}_t} \theta_n^{t+1}$
15:     Global relaxation: $\theta^{t+1} \leftarrow (1-\rho)\theta^t + \rho\tilde{\theta}^{t+1}$
16: **end for**
17: **Return:** $\theta^{T-1}$

---

## 3.2 LOCAL FAST KM GRADIENT CORRECTION

Consider client $n$ at round $t$. In standard FL, the client updates the model directly following its local gradients $g_n(\theta_n^{t,e})$:

$$\theta_n^{t,e+1} = \theta_n^{t,e} - \eta\, g_n(\theta_n^{t,e}), \quad \theta_n^{t,0} := \theta^t.$$

Although this approach works well under homogeneous data, in practice client datasets are often heterogeneous. As a result, each client optimizes its local objective, which may differ significantly

from the global objective. Over multiple local updates, these differences accumulate, causing the client's local trajectory to drift away from the path that would optimally reduce the global loss. This client drift can slow convergence and even destabilize training. To address this issue, we introduce a *correction vector* $\Delta_n^t$ that captures the cumulative historical deviation of the client updates. The correction vector is updated using a fast Krasnosel'skiĭ–Mann (KM) iteration (Boţ & Nguyen, 2023; Sun et al., 2025):

$$\begin{cases} \tilde{\Delta}_n^t = \Delta_n^{t-1} - \beta\big(\theta_n^{t,E-1} - \theta^t\big), & \beta > 0, \\ \Delta_n^t = \frac{2t+\gamma}{2(t+\gamma)}\big(\tilde{\Delta}_n^t + \Delta_n^{t-1}\big) - \frac{t}{t+\gamma}\tilde{\Delta}_n^{t-1}, & \gamma \geq 2. \end{cases}$$

This vector is then used to adjust the local gradient:

$$\tilde{g}_n(\theta_n^{t,e}) = g_n(\theta_n^{t,e}) - \Delta_n^t.$$

After $E$ local epochs, the client sends the updated local model $\theta_n^{t+1} := \theta_n^{t,E-1}$ to the server.

By incorporating a richer history of gradient deviations directly into the local updates, this unified correction mechanism effectively mitigates client drift while implicitly accelerating convergence and improving stability.

### 3.3 GLOBAL RELAXATION ACCELERATION

Once local updates are received, the server first performs standard weighted averaging:

$$\tilde{\theta}^{t+1} = \sum_{n\in\mathcal{C}_t} \omega_n^t\,\theta_n^{t+1}, \quad \omega_n^t \in (0,1), \ \sum_{n\in\mathcal{C}_t} \omega_n^t = 1.$$

For simplicity, we set $\omega_i^t = \omega_j^t = 1/|\mathcal{C}_t|$ for all $i,j \in \mathcal{C}_t$. Instead of immediately broadcasting $\tilde{\theta}^{t+1}$ to clients, we further introduce a relaxation step to accelerate global convergence:

$$\theta^{t+1} = (1-\rho)\theta^t + \rho\tilde{\theta}^{t+1}, \quad 0 < \rho \leq 2.$$

Here, $\rho = 1$ reduces to the aggregation scheme used in FedAvg, $\rho > 1$ accelerates convergence through over-relaxation, while $\rho < 1$ enhances stability. Our ablation studies later demonstrate that this flexibility provides an effective way to speed up global progress.

**Remark 1.** *Setting for hyperparameters.* The hyperparameters for the fast KM iteration and global relaxation acceleration are selected according to the ranges suggested by convex optimization theory (Boţ & Nguyen, 2023; Sun et al., 2025). Since our focus is on more complex federated learning scenarios, these ranges can be slightly relaxed. Nonetheless, empirical evidence suggests that choosing hyperparameters within the theoretically recommended ranges usually yields strong performance.

**Remark 2.** *Discussion on Convergence.* Although the fast KM extrapolation has been theoretically shown to accelerate convergence from $O(1/\sqrt{T})$ to $O(1/T)$ for certain fixed-point problems, providing a general convergence rate guarantee for the proposed federated learning method remains challenging. A more detailed discussion is provided in the appendix.

## 4 EXPERIMENTS

In this section, we present a comprehensive empirical evaluation of the proposed FedRKMGC across various datasets and experimental settings. First, we compare its *convergence performance* with state-of-the-art baselines under different datasets, network architectures, and heterogeneous data distributions. Next, we provide *convergence curve* visualizations to illustrate training dynamics and to highlight the method's communication efficiency. We then conduct an *ablation study* to isolate the contributions of fast Krasnosel'skiĭ–Mann acceleration and global relaxation. Finally, we investigate the sensitivity to key *hyperparameters*, the impact of varying the number of participants and the robustness under different random seeds.

**Datasets and Model Architectures.** Experiments are conducted on two widely used image classification benchmarks, CIFAR-10 and CIFAR-100 (Krizhevsky & Hinton, 2009). For the model backbone, we use ResNet-18 by default, and further evaluate scalability with ResNet-34. Following recent FL practices (Wu et al., 2024; He et al., 2016), the standard batch normalization layers are replaced with static BN to improve training stability in heterogeneous environments.

Table 1: Performance comparison on CIFAR-10/100 under various data distributions.

| Method | IID | | Dir(0.5) | | Dir(0.3) | |
|---|---|---|---|---|---|---|
| | ACCL(%) | ACCG(%) | ACCL(%) | ACCG(%) | ACCL(%) | ACCG(%) |
| | | | CIFAR-10 | | | |
| FedAvg | 90.25(↓2.97) | 89.61(↓3.50) | 87.32(↓2.94) | 85.95(↓3.90) | 88.13(↓1.60) | 86.33(↓2.64) |
| FedProx | 90.21(↓3.01) | 89.50(↓3.61) | 85.42(↓4.84) | 83.54(↓6.31) | 86.12(↓3.61) | 84.23(↓4.74) |
| FedGA | 90.33(↓2.89) | 89.46(↓3.65) | 87.69(↓2.57) | 86.51(↓3.34) | 89.00(↓0.73) | 87.29(↓1.68) |
| SCAFFOLD | 92.29(↓0.93) | 91.93(↓1.18) | 88.59(↓1.67) | 87.18(↓2.67) | 89.31(↓0.42) | 87.27(↓1.70) |
| FedDyn | 91.96(↓1.26) | 91.51(↓1.60) | 90.17(↓0.09) | 89.26(↓0.59) | 88.91(↓0.82) | 88.01(↓0.96) |
| FedADMM | 91.62(↓1.60) | 90.89(↓2.22) | 87.37(↓2.89) | 86.30(↓3.55) | 88.38(↓1.35) | 87.71(↓1.26) |
| **FedRKMGC** | **93.22** | **93.11** | **90.26** | **89.85** | **89.73** | **88.97** |
| | | | CIFAR-100 | | | |
| FedAvg | 56.13(↓14.46) | 53.17(↓15.35) | 56.14(↓12.00) | 53.71(↓12.04) | 55.59(↓10.57) | 52.55(↓12.36) |
| FedProx | 51.73(↓18.43) | 51.73(↓16.79) | 54.10(↓14.02) | 51.64(↓14.11) | 53.74(↓12.42) | 51.64(↓13.27) |
| FedGA | 56.08(↓14.51) | 53.46(↓15.06) | 55.33(↓12.79) | 53.00(↓12.75) | 52.55(↓13.61) | 50.43(↓14.48) |
| SCAFFOLD | 63.81(↓6.78) | 61.81(↓6.71) | 63.56(↓4.56) | 61.01(↓4.74) | 62.32(↓3.84) | 60.12(↓4.79) |
| FedDyn | 64.27(↓6.32) | 62.73(↓5.79) | 64.83(↓3.29) | 62.84(↓2.91) | 64.27(↓1.84) | 62.73(↓2.18) |
| FedADMM | 66.08(↓4.51) | 63.34(↓5.18) | 65.40(↓2.72) | 64.29(↓1.46) | 64.15(↓2.01) | 63.18(↓1.73) |
| **FedRKMGC** | **70.59** | **68.52** | **68.12** | **65.75** | **66.16** | **64.91** |

∗ ACCL and ACCG denote local test accuracy and global test accuracy, respectively.
∗ The numbers in parentheses indicate the difference with our method (FedRKMGC).

**Data Partitioning.** To simulate varying degrees of statistical heterogeneity, we generate client data distributions using a Dirichlet allocation with concentration parameter $\alpha$ (Wu et al., 2024) (denoted as Dir($\alpha$)). Smaller values of $\alpha$ correspond to stronger non-IID conditions.

**Baselines.** We compare FedRKMGC against several representative FL algorithms, including FedAvg (McMahan et al., 2017), FedProx (Li et al., 2020), SCAFFOLD (Karimireddy et al., 2020), FedDyn (Acar et al., 2021), FedGA (Dandi et al., 2022), and FedADMM (Zhou & Li, 2023).

**Training Settings.** Unless otherwise specified, all methods are trained under the same configurations to ensure fairness. Each experiment consists of 1000 communication rounds. In each round, 10 clients are randomly selected from a total of 100 (i.e., 10% participation). The selected clients perform 5 local epochs[1] with a batch size of 20. We adopt stochastic gradient descent as the optimizer, with a learning rate of 0.01 and momentum of 0.8. For FedRKMGC, the gradient correction parameter is set to $\beta = 0.03$, the relaxation parameter to $\rho = 1.5$, and the fast KM parameter to $\gamma = 500$. More detailed hyperparameter settings are provided in Appendix.

**Evaluation Metrics.** We evaluate both global and local test accuracies, where the global accuracy refers to the performance of the global model on the server's test set, whereas local accuracy is defined as the average performance of the global model when evaluated on each client's local test set. To reduce randomness, we report the mean Top-1 accuracy over the final 20 rounds of training.

## 4.1 CONVERGENCE PERFORMANCE COMPARISON

We first compare the convergence performance of different methods across various datasets and data heterogeneity levels. The global and local test accuracies are reported in Table 1, from which we can see that our proposed method consistently outperforms existing approaches, with the advantage becoming more pronounced on more complex datasets. For instance, FedRKMGC achieves a global test accuracy that is 5.18% higher than the second-best method on CIFAR-100 with IID partitioning.

---

[1]For fairness, FedADMM (Zhou & Li, 2023) is also trained with the same number of local epochs as other baselines.

Table 2: Performance comparison on CIFAR-100 with ResNet-34 under various data distributions.

| Method | IID | | Dir(0.5) | | Dir(0.3) | |
|---|---|---|---|---|---|---|
| | ACCL(%) | ACCG(%) | ACCL(%) | ACCG(%) | ACCL(%) | ACCG(%) |
| FedAvg | 54.63(↓15.21) | 52.80(↓14.26) | 55.15(↓12.59) | 52.64(↓12.86) | 55.15(↓10.43) | 51.76(↓11.26) |
| FedProx | 52.37(↓17.47) | 49.74(↓17.32) | 51.56(↓16.18) | 49.57(↓15.93) | 53.65(↓11.93) | 51.37(↓11.65) |
| FedGA | 56.70(↓13.14) | 52.85(↓14.21) | 55.57(↓12.17) | 53.04(↓12.46) | 56.78(↓8.80) | 54.86(↓8.16) |
| SCAFFOLD | 63.23(↓6.61) | 60.67(↓6.39) | 62.53(↓5.21) | 59.29(↓6.21) | 62.21(↓3.37) | 59.28(↓3.74) |
| FedDyn | 65.03(↓4.81) | 62.67(↓4.39) | 64.16(↓3.58) | 62.20(↓3.30) | 63.60(↓1.98) | 62.03(↓0.99) |
| FedADMM | 65.53(↓4.31) | 63.02(↓4.04) | 64.50(↓3.24) | 62.76(↓2.74) | 63.88(↓1.70) | 62.61(↓0.41) |
| **FedRKMGC** | **69.84** | **67.06** | **67.74** | **65.50** | **65.58** | **63.02** |

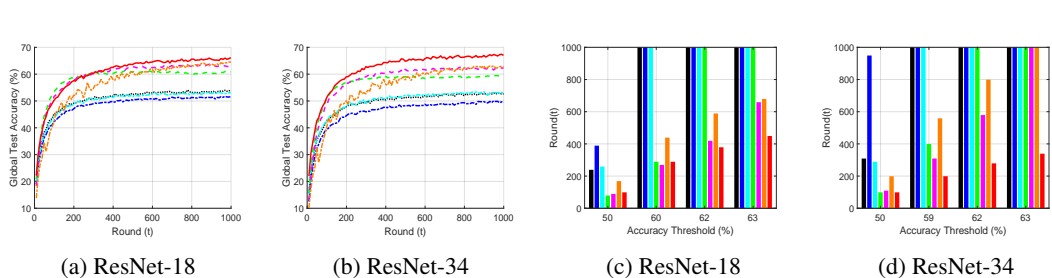

(a) ResNet-18   (b) ResNet-34   (c) ResNet-18   (d) ResNet-34

Figure 2: Convergence curves and communication efficiency comparison on CIFAR-100 with Dir(0.5) under different network architectures.

## 4.2 SCALABILITY TO LARGER NETWORK ARCHITECTURES

To further examine the scalability and generalization ability of our proposed method, we extend the evaluation to a deeper backbone, ResNet-34, on the CIFAR-100 dataset under various data distribution scenarios. The comparison results with other state-of-the-art baselines are summarized in Table 2. From the table, it can be observed that our FedRKMGC still consistently achieves superior convergence performance across all distribution settings, demonstrating its robustness and effectiveness even when applied to larger network architectures.

## 4.3 CONVERGENCE CURVES AND COMMUNICATION EFFICIENCY

To gain deeper insights into the training dynamics, we plot the convergence behavior of different methods on CIFAR-100 with Dir(0.5), using both ResNet-18 and ResNet-34 backbones, as shown in Figure 2. Communication efficiency is evaluated by the number of communication rounds required to reach a target accuracy. From Figures 2(a)-(b), it can be observed that our proposed FedRKMGC not only converges faster but also achieves higher final test accuracy compared with competing baselines. Furthermore, Figures 2(c)-(d) demonstrate that FedRKMGC requires significantly fewer communication rounds to reach the same target accuracy. For instance, with the ResNet-34 backbone, FedRKMGC attains 62% accuracy in only 279 rounds, whereas the second-best method requires 579 rounds, over twice as many communication rounds as ours. Moreover, several competing approaches fail to reach this accuracy within 1000 rounds.

## 4.4 ABLATION STUDY

To further validate the individual contributions of fast KM acceleration and global relaxation, we evaluate the following variants for FedRKMGC:

- **raw (i.e., w/o KM & Re)**: both fast KM and relaxation are removed.
- **w Re**: fast KM is removed, while relaxation is retained ($\rho = 1.5$).
- **w Re & fast KM**: FedRKMGC with fast KM ($\gamma = 500$) and relaxation ($\rho = 1.5$).

Table 3: Ablation study for FedRKMGC on CIFAR-100 under various data distribution scenarios.

| FedRKMGC | IID | | Dir(0.5) | | Dir(0.3) | |
|---|---|---|---|---|---|---|
| | ACCL(%) | ACCG(%) | ACCL(%) | ACCG(%) | ACCL(%) | ACCG(%) |
| raw | 67.74($\downarrow$2.85) | 65.29($\downarrow$3.23) | 64.54($\downarrow$3.58) | 62.39($\downarrow$3.36) | 62.46($\downarrow$3.70) | 60.61($\downarrow$4.30) |
| w Re | 69.00($\downarrow$1.59) | 66.67($\downarrow$1.85) | 66.13($\downarrow$1.99) | 63.79($\downarrow$1.96) | 64.05($\downarrow$2.11) | 62.18($\downarrow$2.73) |
| **w Re & fast KM** | **70.59** | **68.52** | **68.12** | **65.75** | **66.16** | **64.91** |

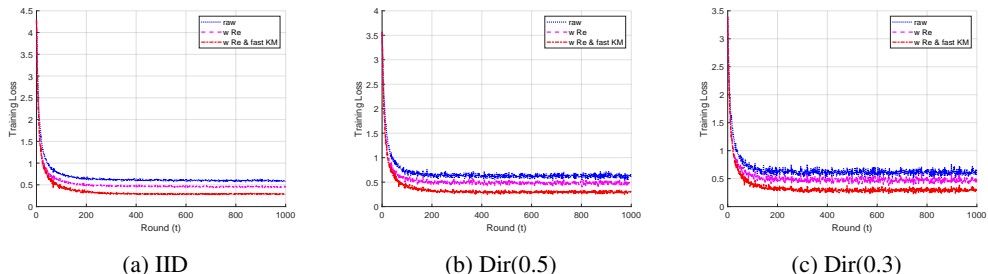

(a) IID        (b) Dir(0.5)        (c) Dir(0.3)

Figure 3: Convergence of training loss on CIFAR-100 under various data distribution scenarios.

We conduct experiments on the CIFAR-100 dataset under different data heterogeneity settings, with results reported in Table 3 and Figure 3. From these results, we observe that comparing **raw** with **w Re** clearly shows the contribution of the relaxation technique in accelerating convergence. Furthermore, comparing **w re** with FedRKMGC (i.e., **w Re & fast KM**) demonstrates that fast KM iteration further enhances convergence speed. These findings highlight the potential of applying classical optimization acceleration techniques in complex, real-world federated learning scenarios. They also motivate the future design of more appropriate acceleration strategies for specific FL settings.

## 4.5 HYPERPARAMETER SENSITIVITY

To further investigate the robustness of our method, we study the impact of key hyperparameters, i.e., the correction parameter $\beta$, the relaxation parameter $\rho$, and the KM parameter $\gamma$, on convergence within a relatively large search space, as illustrated in Figure 4. Although relaxation and KM parameters are introduced to accelerate convergence, the results in Figure 4(b) and Figure 4(c) show that our method is not particularly sensitive to their values, even when explored over wide ranges. In contrast, similar to other gradient correction methods, the correction parameter $\beta$ has a more pronounced influence on convergence performance. However, identifying optimal hyperparameter configurations remains a challenging problem, and we leave more systematic strategies for hyperparameter selection as an important direction for future research.

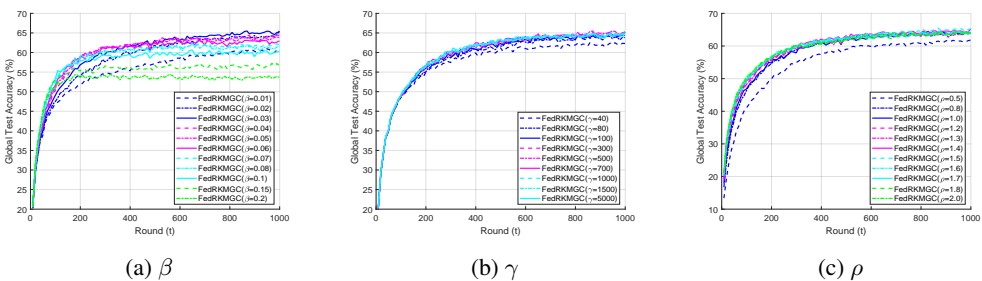

(a) $\beta$        (b) $\gamma$        (c) $\rho$

Figure 4: Impact of hyperparameters in FedRKMGC on CIFAR-100 with Dir(0.3).

Table 4: Performance comparison on CIFAR-100 with Dir(0.3) under varying numbers of clients $N$.

| Method | N=100 | | N=150 | | N=200 | |
|---|---|---|---|---|---|---|
| | ACCL(%) | ACCG(%) | ACCL(%) | ACCG(%) | ACCL(%) | ACCG(%) |
| FedAvg | 55.59(↓10.57) | 52.55(↓12.36) | 52.19(↓14.62) | 50.71(↓14.84) | 49.53(↓16.55) | 48.25(↓17.20) |
| FedProx | 53.74(↓12.42) | 51.64(↓13.27) | 50.60(↓16.21) | 48.59(↓16.96) | 48.79(↓17.29) | 46.98(↓18.47) |
| FedGA | 52.55(↓13.61) | 50.43(↓14.48) | 52.97(↓13.84) | 50.81(↓14.74) | 52.55(↓13.53) | 50.43(↓15.02) |
| SCAFFOLD | 62.32(↓3.84) | 60.12(↓4.79) | 62.32(↓4.49) | 61.04(↓4.51) | 62.64(↓3.44) | 61.55(↓3.90) |
| FedDyn | 64.27(↓1.89) | 62.73(↓2.18) | 62.02(↓4.79) | 59.87(↓5.68) | 61.22(↓4.86) | 59.72(↓5.73) |
| FedADMM | 64.15(↓2.01) | 63.18(↓1.73) | 62.00(↓4.81) | 60.08(↓5.47) | 59.86(↓6.22) | 58.92(↓6.53) |
| **FedRKMGC** | **66.16** | **64.91** | **66.81** | **65.55** | **66.08** | **65.45** |

Table 5: Global test accuracy on CIFAR-100 with Dir(0.3) under different random seeds (S).

| Method | S=42 | S=731 | S=918 | S=1949 | S=2026 | mean | std |
|---|---|---|---|---|---|---|---|
| FedAvg | 52.35 | 52.61 | 51.98 | 52.47 | 54.32 | 52.75 | 0.91 |
| FedProx | 51.93 | 51.87 | 50.50 | 51.21 | 53.29 | 51.76 | 1.03 |
| FedGA | 53.77 | 53.96 | 52.93 | 52.09 | 55.07 | 53.56 | 1.12 |
| SCAFFOLD | 59.84 | 60.22 | 59.64 | 60.77 | 61.58 | 60.41 | 0.78 |
| FedDyn | 62.50 | 62.94 | 61.83 | 61.70 | 63.08 | 62.41 | 0.63 |
| FedADMM | 62.81 | 62.60 | 62.31 | 63.08 | 62.23 | 62.61 | 0.35 |
| **FedRKMGC** | **64.96** | **65.20** | **65.11** | **64.70** | **64.89** | **64.97** | **0.19** |

## 4.6 IMPACT OF CLIENT NUMBER

We further evaluate the robustness of different methods on CIFAR-100 with Dirichlet partitioning ($\alpha = 0.3$) under different numbers of clients, specifically $N = 100, 150, 200$, with a client participation ratio of 10%, as reported in Table 4. The results show that across all these settings, our proposed method consistently achieves the highest global and local test accuracies, demonstrating strong robustness to changes in the number of clients.

## 4.7 RANDOM SEED ROBUSTNESS

To further assess the stability of the proposed method, we repeat key experiments on CIFAR-100 with Dir(0.3) under multiple random seeds and report both the mean and standard deviation of the results in Table 5. From the results, we observe that classical baselines such as FedAvg and FedProx exhibit relatively large performance fluctuations across different seeds, while more advanced methods like SCAFFOLD, FedDyn, and FedADMM achieve improved consistency. In contrast, our proposed FedRKMGC not only consistently outperforms all baselines in terms of average accuracy but also achieves the lowest standard deviation, demonstrating remarkable robustness to random initialization. These findings further highlight the reliability of our approach under practical federated learning scenarios where randomness is inevitable.

## 5 CONCLUSION

In this work, we propose FedRKMGC, a novel federated learning framework that combines gradient correction with fast Krasnosel'skiĭ–Mann acceleration and global relaxation techniques. By integrating these complementary mechanisms, FedRKMGC effectively mitigates client drift caused by heterogeneous data while accelerating convergence, improving both training stability and communication efficiency. Extensive experiments on standard FL benchmarks, including CIFAR-10 and CIFAR-100 under various non-IID settings, demonstrate that FedRKMGC consistently outperforms state-of-the-art methods in terms of convergence speed, final test accuracy, and communication cost. Our work highlights the potential of incorporating classical optimization acceleration strategies into federated learning, offering a promising direction for future research.

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

## A  MOTIVATION FOR FAST KM CORRECTION

A widely acknowledged principle in the design of first-order acceleration schemes is that the momentum or extrapolation term is constructed from historical information, typically previous iterates or directions. While using multi-step histories may potentially yield stronger acceleration, it also introduces higher storage costs and substantially more complexity in designing stable update rules. In contrast, leveraging only the most recent step strikes a favorable balance between computational efficiency, memory cost, and theoretical tractability. For this reason, our method intentionally focuses on designing an acceleration mechanism that relies solely on the previous correction.

More specifically, our extrapolation step is directly inspired by recent advances in optimization theory, where fast KM iteration has been shown to achieve accelerated fixed-point convergence using only one-step historical information. In particular, as shown in (Boţ & Nguyen, 2023), the fast KM update is given by

$$
\begin{aligned}
x^{t+1} &= x^t + \frac{\gamma}{2(t+1+\gamma)}\left(\tilde{x}^{t+1} - x^t\right) + \frac{t+1}{t+1+\gamma}\left(\tilde{x}^{t+1} - \tilde{x}^t\right) \\
&= \frac{2(t+1)+\gamma}{2(t+1+\gamma)}\left(\tilde{x}^{t+1} + x^t\right) - \frac{t+1}{t+1+\gamma}\,\tilde{x}^t,
\end{aligned}
$$

which is a discretization of the dynamical system proposed in (Boţ et al., 2025). Inspired by this design, we propose our FedRKMGC method to accelerate convergence.

## B  COMPLEXITY ANALYSIS

In this section, we compare the computational complexity of SCAFFOLD, FedDyn, and our FedRKMGC.

For all three methods, the dominant computation in each communication round consists mainly of vector additions and vector-scalar multiplications, both of which have a complexity of $\mathcal{O}(p)$ with $p$ being the dimension of $\theta^t$. The detailed per-round complexity of the three algorithms is summarized below. As shown, all methods have the same order of computational complexity.

Table 6: Comparison of computational complexity with SCAFFOLD and FedDyn.

| | Client | | | Server | Total |
|---|---|---|---|---|---|
| | epoch | | other | | |
| | RG | $\theta_n^t$ | | | |
| SCAFFOLD | $C_g$ | $3p$ | $7p$ | $[2p + (|\mathcal{C}_t| - 1)p]$(for $\theta^{t+1}$) $+[2p + (|\mathcal{C}_t| - 1)p]$(for $c^{t+1}$) | $C_gE + [(3E + 9)|\mathcal{C}_t| + 2]p$ $= \mathcal{O}(p)$ |
| FedDyn | $C_g$ | $6p$ | $3p$ | $[2p + (|\mathcal{C}_t| - 1)p + |\mathcal{C}_t|p]$(for $h^{t+1}$) $+[2p + (|\mathcal{C}_t| - 1)p]$(for $\theta^{t+1}$) | $C_gE + [(6E + 6)|\mathcal{C}_t| + 2]p$ $= \mathcal{O}(p)$ |
| FedRKMGC | $C_g$ | $3p$ | $7p$ | $[(|\mathcal{C}_t| - 1)p]$(for $\tilde{\theta}^{t+1}$) $+[3p]$(for $\theta^{t+1}$) | $C_gE + [(3E + 8)|\mathcal{C}_t| + 2]p$ $= \mathcal{O}(p)$ |

Notes: RG denotes raw gradient, $C_g$ denotes the computational cost of evaluating the gradient with respect to $\theta^t$ and $E$ is the number of local epoch.

## C  DISCUSSION ON CONVERGENCE

Our method is inspired by the framework in the recent work (Sun et al., 2025), which studies the following convex optimization problem:

$$
\begin{aligned}
\min_{y \in \mathbb{Y}, z \in \mathbb{Z}} \quad & f_1(y) + f_2(z) \\
\text{s.t.} \quad & B_1 y + B_2 z = c,
\end{aligned}
$$

where $f_1$ and $f_2$ are proper closed convex functions, and $B_1$ and $B_2$ are given linear operators. The iterative scheme in (Sun et al., 2025) is given by

$$\begin{cases} \bar{z}^t = \arg\min_{z\in\mathbb{Z}}\{L_\sigma(y^t, z; x^t) + \frac{1}{2}\|z - z^t\|_2\}, \\ \bar{x}^t = x^t + \sigma(B_1 y^t + B_2 \bar{z}^t - c), \\ \bar{y}^t = \arg\min_{y\in\mathbb{Y}}\{L_\sigma(y, \bar{z}^t; \bar{x}^t) + \frac{1}{2}\|y - y^t\|_2\}, \\ \tilde{w}^{t+1} = (1 - \rho)w^t + \rho\bar{w}^t, \\ w^{t+1} = w^t + \frac{2(t+1)+\gamma}{2(t+1+\gamma)}(\tilde{w}^{t+1} + w^t) - \frac{t+1}{t+1+\gamma}\tilde{w}^t \end{cases}$$

with $w^t = (y^t, z^t, x^t)$ and $L_\sigma = f_1(y) + f_2(z) + x^T(B_1 y + B_2 z - c) + \frac{\sigma}{2}\|B_1 y + B_2 z - c\|^2$. This can be compactly rewritten as

$$\begin{cases} \tilde{w}^{t+1} = (1 - \rho)w^t + \rho\mathcal{T}(w^t), \\ w^{t+1} = w^t + \frac{2(t+1)+\gamma}{2(t+1+\gamma)}(\tilde{w}^{t+1} + w^t) - \frac{t+1}{t+1+\gamma}\tilde{w}^t, \end{cases} \quad (1)$$

where $\mathcal{T}(\cdot)$ is an operator. Similarly, our algorithm can be expressed as

$$\begin{cases} \tilde{w}^{t+1} = \mathrm{diag}(I - \rho_1 I, \ldots, I - \rho_{2N+1}I)w^t + \mathrm{diag}(\rho_1 I, \ldots, \rho_{2N+1}I)\tilde{\mathcal{T}}(w^t), \\ w^{t+1} = \tilde{w}^{t+1} + \mathrm{diag}\big(\underbrace{0, \ldots, 0}_{N+1}, \underbrace{1, \ldots, 1}_{N}\big)\Big(w^t + \frac{2(t+1)+\gamma}{2(t+1+\gamma)}(\tilde{w}^{t+1} + w^t) - \frac{t+1}{t+1+\gamma}\tilde{w}^t - \tilde{w}^{t+1}\Big), \end{cases}$$

$$(2)$$

where $w^t = (\theta_1^t, \ldots, \theta_N^t, \theta^t, \Delta_1^t, \ldots, \Delta_N^t)$, $\rho_{N+1} = \rho$ and $\rho_i = 1$ for $i \neq N+1$ and $\tilde{\mathcal{T}}(\cdot)$ is an operator.

Compared with the iterative scheme (1), our update rule (2) applies relaxation and fast KM-type acceleration selectively to certain variables, a design inspired by the unique structure and constraints of federated learning. This produces different coefficient matrices associated with relaxation and fast KM acceleration compared to the formulation in (Sun et al., 2025). Moreover, our framework involves more than two separable blocks, unlike the two-block structure considered in (Sun et al., 2025). These differences make the theoretical convergence analysis of our algorithm considerably more challenging.

As for future directions, we outline two possible strategies for proving convergence:

- *Extension of operator-based analysis.* One can generalize the proof strategy in Sun et al. (2025) to handle multi-variable separable structures and parameter matrices for relaxation and fast KM updates, instead of scalar parameters, and then apply the corresponding theoretical analysis to the practical problem considered here.

- *Dynamical system perspective.* Note that the iteration (1) can be interpreted as a discretization of a dynamical system (Boţ et al., 2025). Similarly, our iteration in (2) can be analyzed by casting it into a dynamical system framework and extending the convergence results from dynamical systems theory as developed in Boţ et al. (2025).

Both perspectives offer promising directions, though the added complexity of federated learning suggests substantial technical challenges remain.

# D EXPERIMENTAL DETAILS

## D.1 DATA PARTITIONING

In our experiments, we follow the non-IID partitioning procedure used in the implementation of (Wu et al., 2024), which itself builds on the approaches described in (Hsu et al., 2019). Concretely, each client $n$ is assigned a class-probability vector $\mathbf{q}^n \in \mathbb{R}^N$ over the $C$ classes with $q_i^n \geq 0$ and $\|\mathbf{q}\|_1 = 1$, which is drawn from a Dirichlet distribution parameterized by $\alpha\mathbf{p}$, i.e.,

$$\mathbf{q}^{(n)} \sim \mathrm{Dir}(\alpha\mathbf{p}).$$

Here, $\mathbf{p}$ denotes the prior class distribution (we use a uniform prior across all classes) and $\alpha > 0$ is a concentration parameter controlling the heterogeneity among clients. Larger values of $\alpha$ lead to client distributions that more closely approximate the IID setting, whereas smaller $\alpha$ induce highly skewed and thus strongly non-IID client datasets. Given a sampled vector $\mathbf{q}^n$, we allocate training examples to client $n$ proportionally to $\mathbf{q}^n$ so that each client holds the preset number of local samples. As an example, on CIFAR-10, we generate a population of 100 clients with 500 images each (drawn from the 50000-image training set), and the test set distribution matches the uniform prior used for $\mathbf{p}$. All data-partitioning scripts are based on the codebase released by Wu et al. (2024).

## D.2 HYPERPARAMETER SETTINGS

Federated learning algorithms are often sensitive to specific hyperparameters that critically affect their convergence performance. To ensure a fair comparison, we perform grid search for all methods on the CIFAR-100 dataset, where the search ranges are chosen around the optimal values suggested in the original papers. The selected ranges and final choices used in our experiments are summarized in Table 7.

Table 7: Hyperparameter search ranges and selected values.

| Method | Search Range | Selected Value |
|---|---|---|
| FedProx | $\mu \in \{0.01, 0.05, 0.1, 0.3\}$ | $\mu = 0.1$ |
| FedGA | $\beta \in \{0.001, 0.005, 0.01, 0.025, 0.1\}$ | $\beta = 0.025$ |
| STEM | $a \in \{0.1, 0.3, 0.5, 0.7, 0.8, 0.9, 0.95, 0.99\}$ | CIFAR10 : $a = 0.8$, CIFAR100 : $a = 0.9$ |
| FedMoS | $a, \beta \in \{0.1, 0.3, 0.5, 0.7, 0.9, 0.95\}$, | CIFAR10 : $a = 0.9, \beta = 0.9, \mu = 0.001$, |
| | $\mu \in \{0.0001, 0.001, 0.01\}$ | CIFAR100 : $a = 0.95, \beta = 0.3, \mu = 0.001$, |
| FedDyn | $\alpha \in \{0.03, 0.05, 0.1, 0.3, 0.5, 0.8, 1.0\}$ | $\alpha = 0.3$ |
| FedADMM | $\sigma \in \{0.01, 0.1, 0.3, 0.5, 0.8, 1.0\}$ | $\sigma = 0.3$ |
| FedRKMGC | $\beta \in \{0.02, 0.03, 0.05, 0.1, 0.2\}$ | $\beta = 0.03$ |

For our proposed FedRKMGC, we observe that the method is relatively insensitive to the relaxation parameter $\rho$ and the fast KM parameter $\gamma$, as shown in Figure 4. Therefore, we fix them to $\rho = 1.5$ and $\gamma = 500$ throughout all experiments.

## D.3 MORE PARAMETER SENSITIVITY ANALYSIS

The suggested hyperparameter ranges, $\gamma \geq 2$ and $\rho \in (0, 2]$, are motivated by recent theoretical works Boţ & Nguyen (2023); Sun et al. (2025) on the inclusion problem and on two-block separable convex optimization. Since our paper focuses on a more complex federated learning scenario, a wider range of hyperparameters can also be explored in practice. To further validate the robustness of these choices, we conducted additional experiments on CIFAR-100 with ResNet-18 under Dir(0.3) using $\gamma \in \{0.1, 0.5, 1.0, 1.5, 2, 10\}$ and $\rho \in \{2.1, 2.2, 2.4, 2.5, 3.0, 4.0\}$. The convergence curves of FedRKMGC under different hyperparameter settings are plotted in Fig. 5. From these experiments, we observe that when $\gamma$ becomes small (in particular, $\gamma \leq 2$), the convergence performance can deteriorate noticeably. Similarly, when $\rho$ is chosen too large (e.g., $\rho > 2.5$), we also observe a degradation in convergence behavior. Based on these empirical findings, we therefore recommend using moderately large values of $\gamma$ and choosing $\rho$ within the interval $(0, 2]$.

It is well-known that hyperparameters often affect the performance of a method, and tuning them usually requires considerable additional resources. Therefore, designing suitable adaptive or learnable schemes for these hyperparameters is an attractive research direction. However, how to develop simple and effective adaptive or learnable approaches remains a challenge.

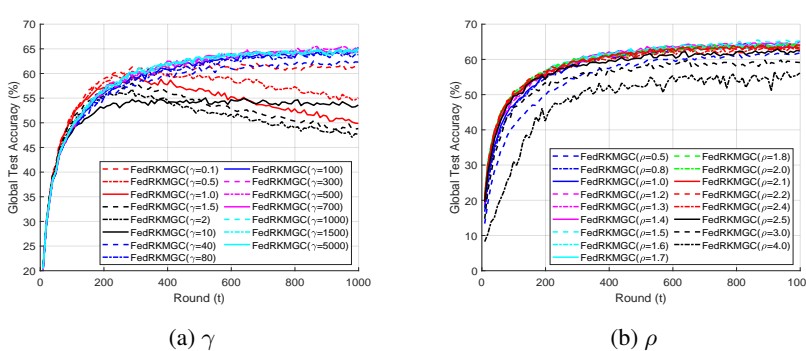

(a) $\gamma$           (b) $\rho$

Figure 5: Impact of hyperparameters in FedRKMGC on CIFAR-100 with Dir(0.3).

Table 8: Performance for FedRKMGC and FedADMM on CIFAR-100 with Dir(0.3) under various local epoch settings.

| ACCG(%) | E=1 | E=3 | E=4 | E=5 | E=6 | E=7 | E=10 |
|---|---|---|---|---|---|---|---|
| FedADMM | 63.08 | 64.26 | 63.48 | 63.18 | 61.88 | 59.95 | 56.10 |
| **FedRKMGC** | **64.15** | **65.28** | **64.65** | **64.91** | **64.22** | **64.43** | **62.87** |

### D.4 EPOCH FOR FEDADMM

For fairness, in our paper we train FedADMM using the same number of local epochs as the other baselines. This follows the common practice adopted in FedDyn, which are also run with SGD-based local updates and an equal number of local epochs to ensure a fair comparison. To further confirm the superiority of our method, we additionally evaluate FedADMM under different choices of local epochs, as shown in the Table 8. The results consistently show that our proposed method maintains a clear advantage across a wide range of epoch settings, which supports the fairness and robustness of our comparisons.

### D.5 COSINE SIMILARITY FOR DIFFERENT METHODS

To provide a more detailed quantitative comparison of different methods, including all ablation variants, we present the per-round average cosine similarity of the participating client models on CIFAR-100 with ResNet-18 under Dir(0.1). The results are plotted in Fig. 6.

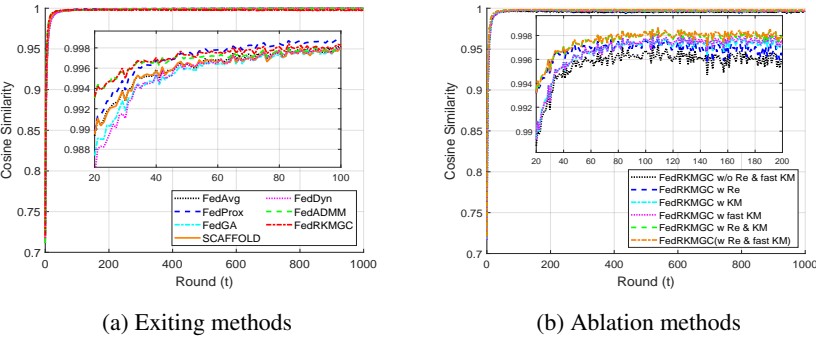

(a) Exiting methods          (b) Ablation methods

Figure 6: Cosine similarity for different methods on CIFAR-100 with ResNet-18 under Dir(0.1).

## D.6 ADDITIONAL ABLATION STUDY

To further disentangle the contribution of each component, we compare the following additional ablation variants for FedRKMGC:

- **raw (i.e., w/o KM & Re)**: both fast KM and relaxation are removed.
- **w Re**: fast KM is removed, while relaxation is retained ($\rho = 1.5$).
- **w KM**: relaxation is removed, while the (standard) KM is retained ($\gamma = 500$).
- **w fast KM**: relaxation is removed, while fast KM is retained ($\gamma = 500$).
- **w Re & KM**: relaxation ($\rho = 1.5$) and the (standard) KM ($\gamma = 500$) are retained.
- **w Re & fast KM (i.e, FedRKMGC)**: full method with fast KM ($\gamma = 500$) and relaxation ($\rho = 1.5$).

Here, the (standard) KM technique corresponds to updating the correction term via (Boţ & Nguyen, 2023)

$$\Delta_n^{t+1} = \frac{2(t+1)+\gamma}{2(t+1+\gamma)}\Delta_n^t + \frac{\gamma}{2(t+1+\gamma)}\tilde{\Delta}_n^{t+1}.$$

In fact, the fast KM iteration can be rewritten as

$$\begin{aligned}
\Delta_n^{t+1} &= \frac{2(t+1)+\gamma}{2(t+1+\gamma)}\left(\tilde{\Delta}_n^{t+1} + \Delta_n^t\right) - \frac{t+1}{t+1+\gamma}\tilde{\Delta}_n^t \\
&= \frac{2(t+1)+\gamma}{2(t+1+\gamma)}\Delta_n^t + \frac{\gamma}{2(t+1+\gamma)}\tilde{\Delta}_n^{t+1} + \frac{t+1}{t+1+\gamma}\left(\tilde{\Delta}_n^{t+1} - \tilde{\Delta}_n^t\right),
\end{aligned}$$

revealing that fast KM can be interpreted as a KM update enhanced with an extrapolation term $\frac{t+1}{t+1+\gamma}\left(\tilde{\Delta}_n^{t+1} - \tilde{\Delta}_n^t\right)$ which proves to have an accelerating effect on the convergence of the fixed point residual as shown in (Boţ & Nguyen, 2023).

The experimental results for CIFAR-100 with ResNet-18 under different data distribution are given in Table 9 and Figure 7, from which we can see that:

- Comparing **raw**, **w Re**, and **w KM**, we observe that both relaxation and KM individually improve convergence speed.
- Comparing **w fast KM** and **w KM**, we find that the extrapolation-enhanced fast KM achieves faster convergence.
- Comparing **w Re & KM**, **w Re & fast KM**, and **w Re**, we observe that incorporating (fast) KM further accelerates training when relaxation is present.

Table 9: Ablation study for FedRKMGC on CIFAR-100 under various data distribution scenarios.

| ACCG(%) | IID | Dir(0.5) | Dir(0.3) | Dir(0.2) | Dir(0.1) | Dir(0.05) |
|---|---|---|---|---|---|---|
| raw | 65.29 ($\downarrow$3.23) | 62.39 ($\downarrow$3.36) | 60.61 ($\downarrow$4.30) | 59.00 ($\downarrow$3.56) | 52.51 ($\downarrow$5.66) | 43.12 ($\downarrow$5.19) |
| w Re | 66.67 ($\downarrow$1.85) | 63.79 ($\downarrow$1.96) | 62.18 ($\downarrow$2.73) | 59.82 ($\downarrow$2.74) | 54.37 ($\downarrow$3.80) | 44.44 ($\downarrow$3.87) |
| w KM | 66.23 ($\downarrow$2.29) | 63.18 ($\downarrow$2.57) | 61.06 ($\downarrow$3.85) | 59.13 ($\downarrow$3.43) | 54.18 ($\downarrow$3.99) | 45.10 ($\downarrow$3.21) |
| w fast KM | **68.68** ($\uparrow$0.16) | **65.93** ($\uparrow$0.18) | 63.85 ($\downarrow$1.06) | 62.37 ($\downarrow$0.19) | 56.85 ($\downarrow$1.32) | 47.59 ($\downarrow$0.72) |
| w Re & KM | 66.41 ($\downarrow$2.11) | 63.33 ($\downarrow$2.42) | 62.43 ($\downarrow$2.48) | 60.34 ($\downarrow$2.22) | 55.70 ($\downarrow$2.47) | 46.05 ($\downarrow$2.26) |
| **w Re & fast KM** | 68.52 | 65.75 | **64.91** | **62.56** | **58.17** | **48.31** |

## D.7 FEDRKMGC UNDER HIGH DATA HETEROGENEITY

To further assess the performance stability and convergence advantages of our method, particularly under high data heterogeneity, we additionally conducted comparative experiments on CIFAR-100

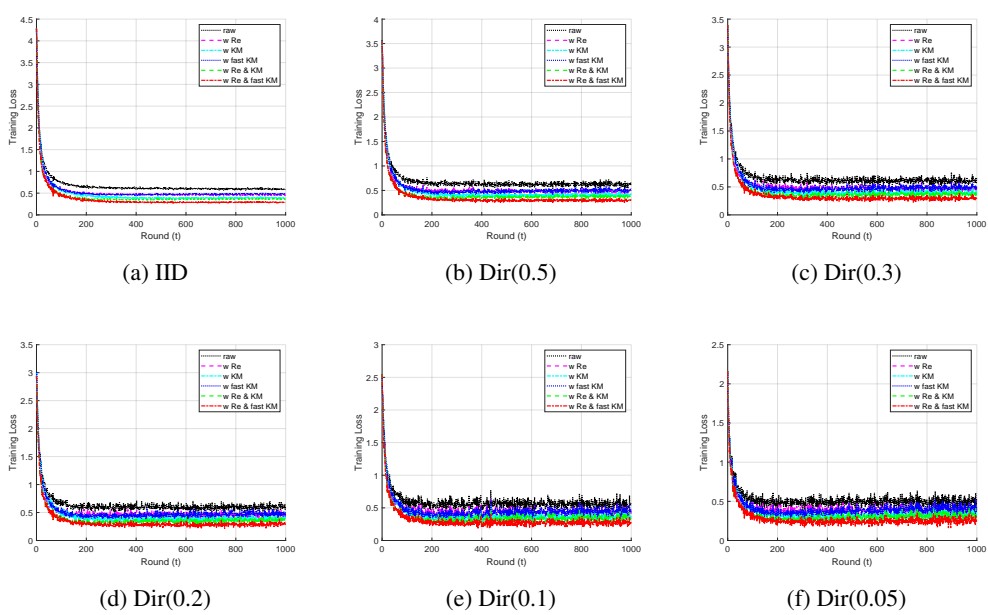

Figure 7: Convergence of training loss on CIFAR-100 under various data distribution scenarios.

with ResNet-18 under Dirichlet partitions of $\alpha = 0.1, 0.2$. The results for different methods under different data heterogeneous settings are summarized in 10. These results show that, even in highly heterogeneous settings, our approach maintains a clear advantage. This further confirms the effectiveness and robustness of our method.

Table 10: Performance comparison on CIFAR-100 with ResNet-18 under various data distributions.

| ACCG(%) | IID | Dir(0.5) | Dir(0.3) | Dir(0.2) | Dir(0.1) |
|---|---|---|---|---|---|
| FedAvg | 53.17 ($\downarrow$15.35) | 53.71 ($\downarrow$12.04) | 52.55 ($\downarrow$12.36) | 52.38 ($\downarrow$10.18) | 48.70 ($\downarrow$9.47) |
| FedProx | 51.73 ($\downarrow$16.79) | 51.64 ($\downarrow$14.11) | 51.64 ($\downarrow$13.27) | 50.86 ($\downarrow$11.70) | 49.33 ($\downarrow$8.84) |
| FedGA | 53.46 ($\downarrow$15.06) | 53.00 ($\downarrow$12.75) | 50.43 ($\downarrow$14.48) | 53.48 ($\downarrow$9.08) | 48.57 ($\downarrow$9.60) |
| SCAFFOLD | 61.81 ($\downarrow$6.71) | 61.01 ($\downarrow$4.74) | 60.12 ($\downarrow$4.79) | 60.45 ($\downarrow$2.11) | 57.80 ($\downarrow$0.37) |
| FedDyn | 62.73 ($\downarrow$5.79) | 62.84 ($\downarrow$2.91) | 62.73 ($\downarrow$2.18) | 61.70 ($\downarrow$0.86) | 57.52 ($\downarrow$0.65) |
| FedADMM | 63.34 ($\downarrow$5.18) | 64.29 ($\downarrow$1.46) | 63.18 ($\downarrow$1.73) | 61.06 ($\downarrow$1.50) | 54.95 ($\downarrow$3.22) |
| **FedRKMGC** | **68.52** | **65.75** | **64.91** | **62.56** | **58.17** |

### D.8 ADDITIONAL EXPERIMENTS ON AGNEWS DATASET

To further demonstrate the generality of our method on non-vision tasks, we conducted additional experiments using different methods on the AgNews dataset with the LSTM model and the CNNL-STM model. The performance comparison for different methods under IID and Dir(1) are presented in Fig. 8. From these results, it can be observed that our method still maintains certain advantages on non-vision tasks, further validating the superiority of our approach.

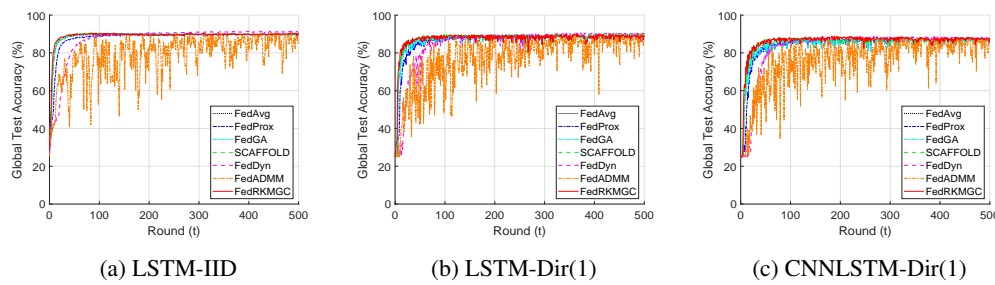

(a) LSTM-IID          (b) LSTM-Dir(1)          (c) CNNLSTM-Dir(1)

Figure 8: Performance comparison for different methods on AgNews with LSTM and CNNLSTM under various data distribution scenarios.

### D.9 COMPARED WITH MORE RECENT NETHODS

#### D.9.1 COMPARED WITH FEDU$^2$ AND FEDREP

FedU$^2$ is designed for federated unsupervised learning, whereas our work focuses on the supervised setting. FedRep targets personalized federated learning, aiming to train a personalized model for each client, while our objective is to learn a single shared global model.

Our contribution lies in accelerating the optimization process via relaxation and fast KM techniques, thereby improving the convergence speed. These techniques are in fact orthogonal to the goals of FedU$^2$ and FedRep. One could try to design new methods for unsupervised or personalized FL frameworks based on our acceleration mechanisms to further improve their efficiency; however, such extensions fall outside the scope of this work and constitute promising directions for future research.

#### D.9.2 COMPARED WITH FEDGA25, STEM, FEDMOS

To further demonstrate the advantages of our method, we additionally compare FedRKMGC with several more recent acceleration techniques. Specifically, we include experiments with FedGA24(Xiao et al., 2024), STEM(Khanduri et al., 2021), and FedMoS(Wang et al., 2023) on CIFAR-10 and CIFAR-100 using ResNet-18 under various data distributions. The results are presented in Table 11. As shown in the table, our method consistently outperforms these recent baselines across all settings, highlighting its robustness and strong acceleration capability.

Table 11: Performance comparison with more recent methods on CIFAR-10 and CIFAR-100 with ResNet-18 under various data distributions.

| ACCG% | CIFAR-10 | | | CIFAR-100 | | |
|---|---|---|---|---|---|---|
| | IID | Dir(0.5) | Dir(0.3) | IID | Dir(0.5) | Dir(0.3) |
| FedAvg | 89.61(↓3.50) | 85.95(↓3.90) | 86.33(↓2.64) | 53.17(↓15.35) | 53.71(↓12.04) | 52.55(↓12.36) |
| FedProx | 89.50(↓3.61) | 83.54(↓6.31) | 84.23(↓4.74) | 51.73(↓16.79) | 51.64(↓14.11) | 51.64(↓13.27) |
| FedGA | 89.46(↓3.65) | 86.51(↓3.34) | 87.29(↓1.68) | 53.46(↓15.06) | 53.00(↓12.75) | 50.43(↓14.48) |
| FedGA24 | 89.65(↓3.46) | 83.84(↓6.01) | 83.34(↓5.63) | 53.43(↓15.09) | 53.04(↓12.71) | 53.67(↓11.24) |
| STEM | 89.89(↓3.22) | 85.74(↓4.11) | 84.96(↓4.01) | 51.90(↓16.62) | 52.99(↓12.76) | 52.81(↓12.10) |
| FedMoS | 86.74(↓6.37) | 86.57(↓3.28) | 88.07(↓0.90) | 50.80(↓17.72) | 52.13(↓13.62) | 52.11(↓12.80) |
| SCAFFOLD | 91.93(↓1.18) | 87.18(↓2.67) | 87.27(↓1.70) | 61.81(↓6.71) | 61.01(↓4.74) | 60.12(↓4.79) |
| FedDyn | 91.51(↓1.60) | 89.26(↓0.59) | 88.01(↓0.96) | 62.73(↓5.79) | 62.84(↓2.91) | 62.73(↓2.18) |
| FedADMM | 90.89(↓2.22) | 86.30(↓3.55) | 87.71(↓1.26) | 63.34(↓5.18) | 64.29(↓1.46) | 63.18(↓1.73) |
| **FedRKMGC** | **93.11** | **89.85** | **88.97** | **68.52** | **65.75** | **64.91** |

