# OpenReview forum: "FedRKMGC: Towards High-Performance Gradient Correction-based Federated Learning via Relaxation and Fast KM Iteration"
_ICLR.cc/2026/Conference — Submitted to ICLR 2026_

### Official Review · Reviewer_BYUp · 2025-10-27

**Soundness:** 2
**Presentation:** 2
**Contribution:** 2
**Rating:** 4
**Confidence:** 2

**Summary:**

This paper introduces FedRKMGC, a federated learning algorithm that addresses the dual challenges of slow convergence and client drift in heterogeneous data settings. The method integrates three key components: (1) gradient correction to mitigate client drift, (2) fast Krasnosel'skiĭ–Mann (KM) iteration for local acceleration, and (3) global relaxation for server-side acceleration.

**Strengths:**

1. The paper presents an interesting combination of classical optimization techniques (fast KM iteration and relaxation) applied to federated learning.
2. FedRKMGC consistently outperforms baselines across all settings, with particularly impressive gains on CIFAR-100.
3. The method demonstrates significant communication savings, requiring roughly half the rounds of competing methods to reach target accuracy thresholds.

**Weaknesses:**

1. While the authors acknowledge this limitation and provide some discussion in the appendix, the absence of convergence guarantees is a significant weakness for a traditional FL paper targeted on the top conference like ICLR.
2. Only image classification tasks (CIFAR-10/100) are evaluated. And no comparison with more recent acceleration methods in FL.
3. While the authors claim robustness to relaxation (ρ) and KM (γ) parameters, the gradient correction parameter (β) appears quite sensitive based on Figure 4(a).
4. The paper doesn't analyze the additional computational cost of the fast KM iteration and correction vector maintenance at the client side, which could be important for resource-constrained devices.
5. Some notation is introduced without clear definition.

**Questions:**

1. The relationship between the "raw correction" and "fast KM correction" in Algorithm 1 (lines 11-12) needs better motivation. Why is this specific form of extrapolation chosen?
2. The paper mentions that FedADMM is trained with the same number of local epochs "for fairness," but this may not be the optimal configuration for that method.
3. No comparison with standard KM iteration (without the "fast" variant) to quantify the specific benefit of the fast KM acceleration.

---

> ### Author Response · Authors · 2025-11-25
> **1/N: Theoretical analysis (W1)**
>
> We are grateful for the reviewer’s insightful questions and constructive feedback. Detailed responses to all points are provided as follows.
>
> We appreciate the reviewer’s emphasis on theoretical guarantees. Our method combines relaxation on the global model with fast KM extrapolation applied only to the correction vectors, a design that differs substantially from existing formulations in [1,2]. Because of this structural mismatch, the available convergence frameworks are very difficult to directly apply to our setting, even under stronger assumptions.
>
> While deriving a full convergence theory for this hybrid mechanism is nontrivial, we have conducted extensive experiments, including ablations, sensitivity analyses, and evaluations under various datasets, model architectures and high data heterogeneity, that consistently verify the stability and effectiveness of our approach. Establishing a corresponding theoretical framework is an important direction for future work, and we appreciate the reviewer’s suggestion.
>
> [1] Defeng Sun, Yancheng Yuan, Guojun Zhang, and Xinyuan Zhao. Accelerating preconditioned admm via degenerate proximal point mappings. SIAM Journal on Optimization, 35(2):1165–1193, 2025.
>
> [2] Radu Ioan Bot¸ and Dang-Khoa Nguyen. Fast Krasnosel’skiı–Mann algorithm with a convergence rate of the fixed point iteration of $o(1/k)$. SIAM Journal on Numerical Analysis, 61(6):2813–2843, 2023.

---

> ### Author Response · Authors · 2025-11-25
> **2/N: Additional experiments on AgNews dataset and comparison with recent methods (W2)**
>
> **1) Additional experiments on AgNews dataset**
>
> To further demonstrate the generality of our method on non-vision tasks, we conducted additional experiments using different methods on the **AgNews dataset with the LSTM model and the CNNLSTM model**. The convergence curves for different methods under IID and Dir(1) are presented in Appendix D.8.
> It can be observed that our method still maintains certain advantages on non-vision tasks, further validating the superiority of our approach.
>
> **2) Compared with more recent methods**
>
> To further demonstrate the advantages of our method, we additionally compare FedRKMGC with several more recent acceleration techniques. Specifically, we include experiments with **FedGA24[1], STEM[2], and FedMoS[3]** on CIFAR-10 and CIFAR-100 using ResNet-18 under various data distributions. The results are presented in the following table.
> As shown in this table, our method consistently outperforms these recent baselines across all settings, highlighting its robustness and strong acceleration capability.
>
> |        | CIFAR-10               |            |            | CIFAR-100              |            |            |
> |--------|------------------------|------------|------------|------------------------|------------|------------|
> | ACCG%  | IID                    | Dir(0.5)   | Dir(0.3)   | IID                    | Dir(0.5)   | Dir(0.3)   |
> | FedAvg | 89.61(↓3.50)          | 85.95(↓3.90)| 86.33(↓2.64)| 53.17(↓15.35)         | 53.71(↓12.04)| 52.55(↓12.36)|
> | FedProx| 89.50(↓3.61)          | 83.54(↓6.31)| 84.23(↓4.74)| 51.73(↓16.79)         | 51.64(↓14.11)| 51.64(↓13.27)|
> | FedGA  | 89.46(↓3.65)          | 86.51(↓3.34)| 87.29(↓1.68)| 53.46(↓15.06)         | 53.00(↓12.75)| 50.43(↓14.48)|
> | FedGA24| 89.65(↓3.46)          | 83.84(↓6.01)| 83.34(↓5.63)| 53.43(↓15.09)         | 53.04(↓12.71)| 53.67(↓11.24)|
> | STEM   | 89.89(↓3.22)          | 85.74(↓4.11)| 84.96(↓4.01)| 51.90(↓16.62)         | 52.99(↓12.76)| 52.81(↓12.10)|
> | FedMoS | 86.74(↓6.37)          | 86.57(↓3.28)| 88.07(↓0.90)| 50.80(↓17.72)      | 52.13(↓13.62)| 52.11(↓12.80)|
> | SCAFFOLD| 91.93(↓1.18)     | 87.18(↓2.67)| 87.27(↓1.70)| 61.81(↓6.71)          | 61.01(↓4.74)| 60.12(↓4.79)|
> | FedDyn | 91.51(↓1.60)          | 89.26(↓0.59)| 88.01(↓0.96)| 62.73(↓5.79)      | 62.84(↓2.91)| 62.73(↓2.18)|
> | FedADMM| 90.89(↓2.22)          | 86.30(↓3.55)| 87.71(↓1.26)| 63.34(↓5.18)      | 64.29(↓1.46)| 63.18(↓1.73)|
> | **FedRKMGC** | **93.11**        | **89.85**  | **88.97**  | **68.52**             | **65.75**  | **64.91**  |
>
> More discussion can refer to Appendix D.9.
>
> [1] Chenguang Xiao, Zheming Zuo, and Shuo Wang. FedGA: Federated learning with gradient alignment for error asymmetry mitigation, 2024.
>
> [2] Prashant Khanduri, Pranay Sharma, Haibo Yang, Mingyi Hong, Jia Liu, Ketan Rajawat, and Pramod Varshney. STEM: A stochastic two-sided momentum algorithm achieving near-optimal sample and communication complexities for federated learning. In Advances in Neural Information Processing Systems, volume 34, pp. 6050–6061, 2021.
>
> [3] Xiong Wang, Yuxin Chen, Yuqing Li, Xiaofei Liao, Hai Jin, and Bo Li. FedMoS: Taming client drift in federated learning with double momentum and adaptive selection. In IEEE Conference on Computer Communications, pp. 1–10, 2023.

---

> ### Author Response · Authors · 2025-11-25
> **3/N: Parameter sensitivity analysis (W3)**
>
> We acknowledge that the hyperparameter $\beta$ has a greater impact on the convergence performance compared with the parameters $\gamma$ and $\rho$ in FedRKMGC. This behavior is consistent with many existing FL methods such as FedProx, FedGA, FedDyn, and FedADMM, whose corresponding hyperparameters are also known to significantly influence convergence.
>
> A deeper investigation into how to **design simple and suitable adaptive or learnable schemes** for selecting such hyperparameters is indeed an appealing research direction. However, it is well known that developing effective hyperparameter adaptation mechanisms is highly challenging, especially in more realistic and complex federated learning scenarios.

---

> ### Author Response · Authors · 2025-11-25
> **4/N: Complexity analysis (W4)**
>
> **Computational complexity**: For all methods compared in this paper, the dominant computation in each communication round consists mainly of vector additions and vector–scalar multiplications, both of which require $\mathcal{O}(p)$ operations, where $p$ is the dimension of $\theta^t$. For simplicity, we provide a detailed complexity comparison of SCAFFOLD, FedDyn, and our FedRKMGC. As shown in the table below, all three methods share the same order of computational complexity. This observation aligns with the motivation behind many recent accelerated first-order methods, which explicitly aim to improve convergence speed without increasing the computational order.
>
> **Memory complexity:** Many acceleration or correction-based FL methods rely on utilizing historical iterates, which naturally introduces additional memory or communication requirements. In federated learning, communication is often the primary bottleneck in practical deployments, and our method is deliberately designed to avoid increasing the per-round communication cost while improving convergence. Although our approach introduces an extra auxiliary vector on each client, such memory overhead is typically acceptable for resource-constrained devices, where the dominant limiting factor is the dynamic computation and storage gradients. Moreover, the baselines FedDyn, FedADMM, and SCAFFOLD also require storing auxiliary variables on the client side. While FedGA does not increase memory usage in client, it introduces additional communication overhead in every round.
>
> Designing accelerated algorithms that fully exploit local computation and memory in resource-constrained settings is a valuable research direction, but it lies beyond the scope of this work. More details are provided in Appendix B.
>
> |          | Client       | Client       | Client | Server                                                                                                        | Total                                              |
> |----------|--------------|--------------|--------|---------------------------------------------------------------------------------------------------------------|----------------------------------------------------|
> |          | each epoch   | each epoch   | other  |                                                                                                               |                                                    |
> |          | raw gradient | $\theta_n^t$ |        |                                                                                                               |                                                    |
> | SCAFFOLD | $C_g$        | 3p           | 7p     | $[2p+(\|\mathcal{C}_t\|-1)p](for\ \theta^{t+1})+[2p+(\|\mathcal{C}_t\|-1)p](for\ c^{t+1})$                    | $C_gE+[(3E+9)\|\mathcal{C}_t\|+2]p=\mathcal{O}(p)$ |
> | FedDyn   | $C_g$        | 6p           | 3p     | $[2p+(\|\mathcal{C}_t\|-1)p+\|\mathcal{C}_t\|p](for\ h^{t+1})+[2p+(\|\mathcal{C}_t\|-1)p](for\ \theta^{t+1})$ | $C_gE+[(6E+6)\|\mathcal{C}_t\|+2]p=\mathcal{O}(p)$ |
> | FedRKMGC | $C_g$        | 3p           | 7p     | $[(\|\mathcal{C}_t\|-1)p](for\ \tilde{\theta}^{t+1})+[3p](for\ \theta^{t+1})$                                 | $C_gE+[(3E+8)\|\mathcal{C}_t\|+2]p=\mathcal{O}(p)$ |
>
> Notes: $C_g$ denotes the computational cost of evaluating the gradient with respect to $\theta^t$ and $E$ is the number of local epoch.

---

> ### Author Response · Authors · 2025-11-25
> **5/N: Notation and concept clarification (W5)**
>
> We have carefully reviewed the entire manuscript and revised the presentation to ensure that all notation is clearly and consistently defined upon first appearance.

---

> ### Author Response · Authors · 2025-11-25
> **6/N: Motivation for fast KM correction (Q1)**
>
> A widely acknowledged principle in the design of first-order acceleration schemes is that the momentum or extrapolation term is constructed from historical information, typically previous iterates or directions. While using multi-step histories may potentially yield stronger acceleration, it also introduces higher storage costs and substantially more complexity in designing stable update rules. In contrast, leveraging only the most recent step strikes a favorable balance between computational efficiency, memory cost, and theoretical tractability. For this reason, our method intentionally focuses on designing an acceleration mechanism that relies solely on the previous correction.
>
> More specifically, our extrapolation step is directly inspired by recent advances in optimization theory, where fast KM iteration has been shown to achieve accelerated fixed-point convergence using only one-step historical information. In particular, as shown in [1], the fast KM update is given by
>
> $x^{t+1}= x^t + \tfrac{\gamma}{2(t+1+\gamma)} \big(\tilde{x}^{t+1} - x^t) + \tfrac{t+1}{t+1+\gamma} \big(\tilde{x}^{t+1} - \tilde{x}^t \big)=\tfrac{2(t+1)+\gamma}{2(t+1+\gamma)} \big(\tilde{x}^{t+1}+x^t\big) - \tfrac{t+1}{t+1+\gamma}\, \tilde{x}^t,$
>
> which is a discretization of the dynamical system proposed in [2]. Inspired by this design, we propose our FedRKMGC method to accelerate convergence.
> More details are presented in Appendix A.
>
> [1] Radu Ioan Bot¸ and Dang-Khoa Nguyen. Fast Krasnosel’skiı–Mann algorithm with a convergence rate of the fixed point iteration of $o(1/k)$. SIAM Journal on Numerical Analysis, 61(6):2813–2843, 2023.
>
> [2] Radu Ioan Bot¸, Erno Robert Csetnek, and Dang-Khoa Nguyen. Fast optimistic gradient descent ascent (OGDA) method in continuous and discrete time. Foundations of Computational Mathematics, 25(1):163–222, 2025.

---

> ### Author Response · Authors · 2025-11-25
> **7/N: Epoch for FedADMM (Q2)**
>
> For fairness, in our paper we train FedADMM using the same number of local epochs as the other baselines. This follows the common practice adopted in FedDyn, which are also run with SGD-based local updates and an equal number of local epochs to ensure a fair comparison.
>
> To further address the reviewer’s concern, we additionally evaluate FedADMM under different choices of local epochs (as shown in the following table). The results consistently show that our proposed method maintains a clear advantage across a wide range of epoch settings, which supports the fairness and robustness of our comparisons. More details are given in Appendix D.4.
>
> | ACCG(%)     | E=1     | E=3     | E=4     | E=5     | E=6     | E=7     | E=10    |
> |-------------|---------|---------|---------|---------|---------|---------|---------|
> | FedADMM     | 63.08   | 64.26   | 63.48   | 63.18   | 61.88   | 59.95   | 56.10   |
> | **FedRKMGC**| **64.15** | **65.28** | **64.65** | **64.91** | **64.22** | **64.43** | **62.87** |

---

> ### Author Response · Authors · 2025-11-25
> **8/N: Compared with standard KM (Q3)**
>
> The fast KM update is given by
>
> $x^{t+1}= x^t + \tfrac{\gamma}{2(t+1+\gamma)} \big(\tilde{x}^{t+1} - x^t) + \tfrac{t+1}{t+1+\gamma} \big(\tilde{x}^{t+1} - \tilde{x}^t \big),$
>
> which can be viewed as a standard KM iteration augmented by an extrapolation term $\tfrac{t+1}{t+1+\gamma} \big(\tilde{x}^{t+1} - \tilde{x}^t \big).$ To isolate and quantify the benefit of this acceleration component, we additionally compare our method with the standard KM iteration, given by
>
> $x^{t+1}= x^t + \tfrac{\gamma}{2(t+1+\gamma)} \big(\tilde{x}^{t+1} - x^t),$
>
> i.e., the fast KM update without the extrapolation term.
> More specifically, we perform an ablation study in which we progressively remove (i) relaxation, (ii) KM correction, and (iii) fast KM acceleration.
> In particular, we conduct the following ablation study for CIFAR-100 with ResNet-18 under different data distribution:
>
> - **raw (i.e., w/o KM & Re)**: both fast KM and relaxation are removed.
>
> - **w Re (i.e., w/o KM)**: fast KM is removed, while relaxation is retained ($\rho=1.5$).
>
> - **w KM**: relaxation is removed, while the (standard) KM is retained ($\gamma=500$).
>
> - **w fast KM**: relaxation is removed, while fast KM is retained ($\gamma=500$).
>
> - **w Re & KM**: relaxation ($\rho=1.5$) and the (standard) KM ($\gamma=500$) are retained.
>
> - **w Re & fast KM (i.e, FedRKMGC)**: full method with fast KM ($\gamma=500$) and relaxation ($\rho=1.5$).
>
> From the following table, we can see that:
>
> - Comparing **raw**, **w Re**, and **w KM**, we observe that both relaxation and KM individually improve convergence speed.
>
> - Comparing **w fast KM** and **w KM**, we find that the extrapolation-enhanced fast KM achieves faster convergence.
>
> - Comparing **w Re \& KM**, **w Re \& fast KM**, and **w Re**, we observe that incorporating (fast) KM further accelerates training when relaxation is present.
>
> Additional details and extended experimental results have been included in Appendix D.6.
>
>
>
> | ACCG(%)            | IID                     | Dir(0.5)                 | Dir(0.3)                 | Dir(0.2)                 | Dir(0.1)                 | Dir(0.05)                |
> |--------------------|-------------------------|---------------------------|---------------------------|---------------------------|---------------------------|---------------------------|
> | raw                | 65.29 (↓3.23)           | 62.39 (↓3.36)            | 60.61 (↓4.30)            | 58.38 (↓4.18)            | 52.59 (↓5.58)            | 42.45 (↓5.86)            |
> | w Re               | 66.67 (↓1.85)           | 63.79 (↓1.96)            | 62.18 (↓2.73)            | 60.69 (↓1.87)            | 54.40 (↓3.77)            | 44.46 (↓3.85)            |
> | w KM               | 66.23 (↓2.29)           | 63.18 (↓2.57)            | 61.06 (↓3.85)            | 59.13 (↓3.43)            | 54.18 (↓3.99)            | 45.10 (↓3.21)            |
> | w fast KM      | **68.68 (↑0.16)**       | **65.93 (↑0.18)**        | _63.85 (↓1.06)_          | _62.37 (↓0.19)_          | _56.85 (↓1.32)_          | _47.59 (↓0.72)_          |
> | w Re & KM          | 66.41 (↓2.11)           | 63.33 (↓2.42)            | 62.43 (↓2.48)            | 60.34 (↓2.22)            | 55.70 (↓2.47)            | 46.05 (↓2.26)            |
> | **w Re & fast KM** | _68.52_                 | _65.75_                  | **64.91**                | **62.56**                | **58.17**                | **48.31**                |

---

> ### Author Response · Authors · 2025-11-28
> **Summary**
>
> We are grateful to the reviewer for the careful reading and highly valuable comments. In response, we have incorporated the following additions:
>
> - Additional theoretical discussion (W1)
>
> - Experiments on non-vision tasks (W2) - Appendix D.8
>
> - Comparison with recent methods (W2) - Appendix D.9.
>
> - Parameter sensitivity analysis (W3) - Appendix D.3
>
> - Complexity analysis (W4) - Appendix B
>
> - Notation and concept clarification (W5)
>
> - Motivation for fast KM correction (Q1) - Appendix A
>
> - Study on the choice of local epochs for FedADMM (Q2) - Appendix D.4
>
> - Comparison with standard KM (Q3) - Appendix D.6
>
> Additionally, incorporating feedback from other reviewers, we further added:
>
> - More experimental details - Appendix D.1
>
> - Extended ablation studies for FedRKMGC - Appendix D.6
>
> - Experiments under high data heterogeneity - Appendix D.7
>
> - Inter-client cosine similarity analysis - Appendix D.5
>
> Thank you again for your thoughtful review and constructive feedback.

---

### Official Review · Reviewer_2aiZ · 2025-10-31

**Soundness:** 3
**Presentation:** 3
**Contribution:** 3
**Rating:** 6
**Confidence:** 3

**Summary:**

This paper proposes FedRKMGC, a novel federated learning (FL) framework that integrates gradient correction with the fast KM acceleration method and global relaxation technique. It aims to address the problems of slow convergence and client drift in FL under heterogeneous data distributions. The key contributions include: 1) a unified framework that combines gradient correction with fixed-point acceleration to enhance both stability and convergence speed; 2) a two-level acceleration mechanism, with fast KM extrapolation for client-side local updates and global relaxation for server-side aggregation; 3) extensive experiments on CIFAR-10 and CIFAR-100 datasets, demonstrating that FedRKMGC outperforms state-of-the-art FL methods in convergence speed, final accuracy, and communication efficiency.

**Strengths:**

- Originality: The first to combine fast KM acceleration and global relaxation into a unified FL framework.
- Technical depth: Solid grounding in convex optimization and operator theory, connecting FL to fixed-point iteration literature.
- Empirical validation: Extensive experiments on CIFAR-10/100 with multiple non-IID settings, ablations, sensitivity studies, and robustness tests.
- Significance: Improves both stability (drift reduction) and communication efficiency—a central issue in FL.
- Clarity: Strong writing quality, comprehensive experimental section, and thoughtful discussion on future theoretical analysis.

**Weaknesses:**

- Insufficient theoretical analysis: The paper fails to provide a formal theoretical proof of the convergence rate. Although it mentions that fast KM can accelerate convergence from $O(1/\sqrt{T})$ to $O(1/T)$ for fixed-point problems, it does not extend this to the federated learning scenario, leaving the theoretical validity of FedRKMGC incompletely justified.
- Limited hyperparameter guidance: While the paper reports hyperparameter values used in experiments, it lacks a systematic strategy for hyperparameter selection. The sensitivity analysis shows that the correction parameter \(\beta\) significantly impacts performance, but no method is proposed to optimize its value adaptively.
- Narrow dataset coverage: Experiments are only conducted on image classification datasets (CIFAR-10/100). The performance of FedRKMGC on other types of data (e.g., text, tabular) or more complex FL scenarios (e.g., model heterogeneity, non-convex objectives) is untested, limiting the generalizability of the results.

**Questions:**

1. Can the authors quantify the computational overhead of fast KM extrapolation at each client compared to FedDyn or SCAFFOLD?
2. How sensitive is the performance to incorrect tuning of $\gamma$ or $\rho$ beyond the reported ranges? Could adaptive or learnable schemes for these hyperparameters further improve stability?
3. Have the authors explored the applicability to non-vision tasks, e.g., language or sensor data, to test generality?
4. Would it be possible to derive a partial convergence guarantee (e.g., for convex objectives or bounded variance assumptions) to strengthen the theoretical contribution?

---

> ### Author Response · Authors · 2025-11-24
> **1/N: Computation complexity (Q1)**
>
> We thank the reviewer for the insightful questions and valuable suggestions. We address each point in detail below.
>
> For SCAFFOLD, FedDyn, and our FedRKMGC, the dominant computation in each communication round consists mainly of vector additions and vector-scalar multiplications, both of which have a complexity of $\mathcal{O}(p)$ with $p$ being the dimension
> of $\theta^t$. The detailed per-round complexity of the three algorithms is summarized below. As shown, all methods have the same order of computational complexity. Additional details are provided in Appendix B.
> |          | Client       | Client       | Client | Server                                                                                                        | Total                                              |
> |----------|--------------|--------------|--------|---------------------------------------------------------------------------------------------------------------|----------------------------------------------------|
> |          | each epoch   | each epoch   | other  |                                                                                                               |                                                    |
> |          | raw gradient | $\theta_n^t$ |        |                                                                                                               |                                                    |
> | SCAFFOLD | $C_g$        | 3p           | 7p     | $[2p+(\|\mathcal{C}_t\|-1)p](for\ \theta^{t+1})+[2p+(\|\mathcal{C}_t\|-1)p](for\ c^{t+1})$                    | $C_gE+[(3E+9)\|\mathcal{C}_t\|+2]p=\mathcal{O}(p)$ |
> | FedDyn   | $C_g$        | 6p           | 3p     | $[2p+(\|\mathcal{C}_t\|-1)p+\|\mathcal{C}_t\|p](for\ h^{t+1})+[2p+(\|\mathcal{C}_t\|-1)p](for\ \theta^{t+1})$ | $C_gE+[(6E+6)\|\mathcal{C}_t\|+2]p=\mathcal{O}(p)$ |
> | FedRKMGC | $C_g$        | 3p           | 7p     | $[(\|\mathcal{C}_t\|-1)p](for\ \tilde{\theta}^{t+1})+[3p](for\ \theta^{t+1})$                                 | $C_gE+[(3E+8)\|\mathcal{C}_t\|+2]p=\mathcal{O}(p)$ |
>
> Notes: $C_g$ denotes the computational cost of evaluating the gradient with respect to $\theta^t$ and $E$ is the number of local epoch.

---

> ### Author Response · Authors · 2025-11-24
> **2/N: Parameter sensitivity analysis (Q2 & W2)**
>
> he suggested hyperparameter ranges, $\gamma \ge 2$ and $\rho \in (0,2]$, are motivated by recent theoretical works [1,2] on the inclusion problem and on two-block separable convex optimization. Since our paper focuses on a more complex federated learning scenario, a wider range of hyperparameters can also be explored in practice.
> To further validate the robustness of these choices, we conducted additional experiments on CIFAR-100 with ResNet-18 under Dir(0.3) using
> $\gamma=0.1, 0.5, 1.0, 1.5, 2, 10$
> and
> $\rho=2.1, 2.2, 2.4, 2.5, 3.0, 4.0$.
> The convergence curves of FedRKMGC under different hyperparameter settings are presented in Appendix D.3.
> From these experiments, we observe that when $\gamma$ becomes small (in particular, $\gamma \le 2$), the convergence performance can deteriorate noticeably. Similarly, when $\rho$ is chosen too large (e.g., $\rho > 2.5$), we also observe a degradation in convergence behavior. Based on these empirical findings, we therefore recommend using moderately large values of $\gamma$ and choosing $\rho$ within the interval $(0,2]$.
>
> It is well-known that hyperparameters often affect the performance of a method, and tuning them usually requires considerable additional resources. Therefore, designing suitable adaptive or learnable schemes for these hyperparameters is an attractive research direction. However, how to develop simple and effective adaptive or learnable approaches remains a challenge.
>
> [1] Defeng Sun, Yancheng Yuan, Guojun Zhang, and Xinyuan Zhao. Accelerating preconditioned admm via degenerate proximal point mappings. SIAM Journal on Optimization, 35(2):1165–1193, 2025.
>
> [2] Radu Ioan Bot¸ and Dang-Khoa Nguyen. Fast Krasnosel’skiı–Mann algorithm with a convergence rate of the fixed point iteration of $o(1/k)$. SIAM Journal on Numerical Analysis, 61(6):2813–2843, 2023.

---

> ### Author Response · Authors · 2025-11-24
> **3/N: Additional experiments on AgNews dataset (Q3 & W3)**
>
> To further demonstrate the generality of our method on non-vision tasks, we conducted additional experiments using different methods on the **AgNews dataset with the LSTM model and the CNNLSTM model**. The convergence curves for different methods under IID and Dir(1) are presented in Appendix D.8.
> It can be observed that our method still maintains certain advantages on non-vision tasks, further validating the superiority of our approach.

---

> ### Author Response · Authors · 2025-11-24
> **4/N: Theoretical analysis (Q4 & W1)**
>
> We appreciate the reviewer’s suggestion regarding convergence analysis. Indeed, integrating relaxation and fast KM acceleration is a relatively new idea in convex optimization, and our work explores how such techniques can be adapted to the substantially more complex federated learning setting. Importantly, for practical FL considerations, such as limiting local memory usage and ensuring algorithmic efficiency, we apply fast KM only to the correction vectors and apply relaxation solely to the global updates. This design choice makes our method structurally quite different from existing works [1,2], and therefore the existing theoretical frameworks are very hard to be directly applied to our setting.
> Because of these differences, developing a suitable convergence theory for our hybrid scheme is highly nontrivial and would likely require new analytical tools beyond applying stronger assumptions such as convexity or bounded variance. We view this as an interesting and valuable direction for future work.
>
> Despite the theoretical challenges, we have conducted extensive empirical evaluations demonstrating the stability, robustness, and effectiveness of our algorithm across various datasets, architectures, and heterogeneity levels. We believe these results provide strong evidence for the practical merit of our approach and lay the groundwork for future theoretical investigation.
>
> [1] Defeng Sun, Yancheng Yuan, Guojun Zhang, and Xinyuan Zhao. Accelerating preconditioned admm via degenerate proximal point mappings. SIAM Journal on Optimization, 35(2):1165–1193, 2025.
>
> [2] Radu Ioan Bot¸ and Dang-Khoa Nguyen. Fast Krasnosel’skiı–Mann algorithm with a convergence rate of the fixed point iteration of $o(1/k)$. SIAM Journal on Numerical Analysis, 61(6):2813–2843, 2023.

---

> ### Author Response · Authors · 2025-11-28
> **Summary**
>
> We sincerely appreciate the reviewer for the thoughtful comments and helpful suggestions. In response, we have added:
>
> - Computation complexity analysis (Q1) - Appendix B
>
> - Parameter sensitivity analysis (Q2 & W2) - Appendix D.3
>
> - Experiments on non-vision tasks (Q3 & W3) - Appendix D.8
>
> - Theoretical discussion (Q4 & W1)
>
> Furthermore, following suggestions from other reviewers, we additionally incorporated:
>
> - More experimental details - Appendix D.1
>
> - Extended ablations for FedRKMGC - Appendix D.6
>
> - Comparisons with more recent baselines - Appendix D.9
>
> - Experiments under higher data heterogeneity - Appendix D.7
>
> - Inter-client cosine similarity analysis - Appendix D.5
>
> Thank you very much for your constructive feedback and valuable time.

---

### Official Review · Reviewer_hRr7 · 2025-10-31

**Soundness:** 2
**Presentation:** 3
**Contribution:** 2
**Rating:** 6
**Confidence:** 4

**Summary:**

The paper introduces FedRKMGC, a federated learning framework combining gradient correction, fast KM acceleration, and global relaxation to improve convergence and communication efficiency under data heterogeneity. Experiments on CIFAR-10/100 show faster convergence and higher accuracy than state-of-the-art FL methods.

**Strengths:**

The paper presents a creative idea by integrating fast KM acceleration and relaxation into federated learning, showing moderate improvements in convergence and communication efficiency. While not groundbreaking, the approach is well-motivated, and experiments on standard benchmarks demonstrate consistent, if modest, gains over existing methods.

**Weaknesses:**

1. FedRKMGC introduces relation kernelized multi-graph collaboration with KM-based acceleration for federated optimization under non-IID settings. The concept is interesting but closely related to SCAFFOLD (ICML 2020), FedDyn (ICLR 2021), and FedADMM (TPAMI 2023). Including recent methods such as FedU² (CVPR 2024) [1] would clarify novelty.
2. The related work section omits recent multimodal and representation-based FL methods like FedRep [2] and FedU². Broader comparisons would strengthen the positioning.
3. The experimental results are promising but require more details on non-IID splits, client counts, and communication rounds.
4. Add experimental results comparing FedRKMGC with FedDyn, FedRep, and FedU² under identical conditions (e.g., Dirichlet α = 0.1, 0.2, 0.5 with non-IID data splits). Highlight the performance stability and convergence benefits of FedRKMGC, especially under high data heterogeneity.
5. Ablation is limited. Independent evaluation of kernelization, KM acceleration, and relaxation parameters would clarify their contributions.
6. Include an ablation study isolating the RKM module to demonstrate its specific contribution. Analyze inter-client feature alignment (e.g., cosine similarity before and after aggregation) and present


[1] Liao, X., Liu, W., Chen, C., Zhou, P., Yu, F., Zhu, H., Yao, B., Wang, T., Zheng, X., & Tan, Y. (2024). Rethinking the Representation in Federated Unsupervised Learning with Non-IID Data (FedU²). In Proceedings of the IEEE/CVF Conference on Computer Vision and Pattern Recognition (CVPR 2024), pp. 25189–25198.
[2] Collins, L., Hassani, H., Mokhtari, A., & Shakkottai, S. (2021). Exploiting Shared Representations for Personalized Federated Learning. In Proceedings of the 38th International Conference on Machine Learning (ICML 2021), PMLR, pp. 2089–2099.

**Questions:**

1. The experimental results are promising but require more details on non-IID splits, client counts, and communication rounds.
2. Add experimental results comparing FedRKMGC with FedDyn, FedRep, and FedU² under identical conditions (e.g., Dirichlet α = 0.1, 0.2, 0.5 with non-IID data splits). Highlight the performance stability and convergence benefits of FedRKMGC, especially under high data heterogeneity.
3. Ablation is limited. Independent evaluation of kernelization, KM acceleration, and relaxation parameters would clarify their contributions.
4. Include an ablation study isolating the RKM module to demonstrate its specific contribution. Analyze inter-client feature alignment (e.g., cosine similarity before and after aggregation) and present.

---

> ### Author Response · Authors · 2025-11-24
> **1/N: Experimental details (Q1 & W3)**
>
> We appreciate the reviewer’s insightful comments and constructive suggestions. Our detailed responses to each point are provided below.
>
> **Non-IID splits:**
> In our experiments, the construction of non-IID client data follows the implementation released by Wu et al. [1], which is based on the Dirichlet-sampling approaches described in [2]. Concretely, for each client we sample a class-probability vector **q**$\in\mathbb{R}^N$ from a Dirichlet distribution **q**$\sim$Dir($\alpha$**p**), where **p** denotes the prior class distribution over the C classes and $\alpha>0$ is a concentration parameter that controls the degree of heterogeneity across clients. Larger values of $\alpha$ lead to client distributions that more closely approximate the IID setting, whereas smaller $\alpha$ induce highly skewed and thus strongly non-IID client datasets. Given a sampled **q**, each client receives a fixed number of local examples allocated proportionally to its class probabilities.
> As an example, on CIFAR-10, we generate a population of 100 clients with 500 training images per client. The prior **p** is set to the uniform distribution over the 10 classes (consistent with the test set) and client class counts are determined according to the sampled Dirichlet vectors. All data-partitioning scripts are based on the codebase released by Wu et al. [1].
>
> **Client counts & communication rounds:** The details regarding the number of clients and communication rounds are already provided in the **Training Settings** section of the paper. Specifically, each experiment is run for 1000 communication rounds. In each round, 10 clients are randomly selected from a total of 100 (i.e., a 10\% participation rate). More details can be seen in Appendix D.1.
>
> [1] Feijie Wu, Xingchen Wang, Yaqing Wang, Tianci Liu, Lu Su, and Jing Gao. FIARSE: Model-heterogeneous federated learning via importance-aware submodel extraction. In Proceedings of the International Conference on Neural Information Processing Systems, volume 37, pp. 115615–115651, 2024.
>
> [2] Tzu-Ming Harry Hsu, Hang Qi, and Matthew Brown. Measuring the effects of non-identical data distribution for federated visual classification, 2019.

---

> ### Author Response · Authors · 2025-11-24
> **2/N: Ablation for FedRKMGC (Q3, Q4, W5 & W6)**
>
> We have already included an ablation study on the relaxation and fast KM techniques of FedRKMGC in the manuscript, where we compare:
>
> - **raw (i.e., w/o KM & Re)**: both fast KM and relaxation are removed.
>
> - **w Re (i.e., w/o KM)**: fast KM is removed, while relaxation is retained ($\rho=1.5$).
>
> - **w Re & fast KM (i.e, FedRKMGC)**: full method with fast KM ($\gamma=500$) and relaxation ($\rho=1.5$).
>
> To further disentangle the contribution of each component, we add the following additional ablation variants in the revised manuscript for CIFAR-100 with ResNet-18 under different data distribution:
>
> - **w KM**: relaxation is removed, while the (standard) KM is retained ($\gamma=500$).
>
> - **w fast KM**: relaxation is removed, while fast KM is retained ($\gamma=500$).
>
> - **w Re & KM**: relaxation ($\rho=1.5$) and the (standard) KM ($\gamma=500$) are retained.
>
> Here, the (standard) KM technique (requested by Reviewer BYUp) corresponds to updating the correction term via [1]
>
> $$\Delta_n^{t+1}= \big(1-\tfrac{\gamma}{2(t+1+\gamma)} \big)\Delta_n^t +   \tfrac{\gamma}{2(t+1+\gamma)}\tilde{\Delta}_n^{t+1}. $$
>
> In fact, the fast KM iteration can be rewritten as
>
> $$\Delta_n^{t+1}= \tfrac{2(t+1)+\gamma}{2(t+1+\gamma)} \big(\tilde{\Delta}_n^{t+1}+\Delta_n^t\big) - \tfrac{t+1}{t+1+\gamma}\, \tilde{\Delta}_n^t = \big(1-\tfrac{\gamma}{2(t+1+\gamma)} \big)\Delta_n^t +   \tfrac{\gamma}{2(t+1+\gamma)}\tilde{\Delta}_n^{t+1} + \tfrac{t+1}{t+1+\gamma} \big(\tilde{\Delta}_n^{t+1} - \tilde{\Delta}_n^t \big),$$
>
> revealing that fast KM can be interpreted as a KM update enhanced with an extrapolation term $\tfrac{t+1}{t+1+\gamma} \big(\tilde{\Delta}_n^{t+1} - \tilde{\Delta}_n^t \big)$ which has an accelerating effect on the convergence of the fixed point residual as shown in [1].
>
>
> Experimental observations from the following table:
>
> - Comparing **raw**, **w Re**, and **w KM**, we observe that both relaxation and KM individually improve convergence speed.
>
> - Comparing **w fast KM** and **w KM**, we find that the extrapolation-enhanced fast KM achieves faster convergence.
>
> - Comparing **w Re \& KM**, **w Re \& fast KM**, and **w Re**, we observe that incorporating (fast) KM further accelerates training when relaxation is present.
>
> Additional details and extended experimental results have been included in Appendix D.6.
> We hope this clarification addresses your concern. Thank you again for the helpful comment.
>
>
>
> | **ACCG(%)** | IID | Dir(0.5) | Dir(0.3) | Dir(0.2) | Dir(0.1) | Dir(0.05) |
> |------------|-----|----------|----------|----------|----------|-----------|
> | **raw** | 65.29 (↓3.23) | 62.39 (↓3.36) | 60.61 (↓4.30) | 59.00 (↓3.56) | 52.51 (↓5.66) | 43.12 (↓5.19) |
> | **w Re** | 66.67 (↓1.85) | 63.79 (↓1.96) | 62.18 (↓2.73) | 59.82 (↓2.74) | 54.37 (↓3.80) | 44.44 (↓3.87) |
> | **w KM** | 66.23 (↓2.29) | 63.18 (↓2.57) | 61.06 (↓3.85) | 59.13 (↓3.43) | 54.18 (↓3.99) | 45.10 (↓3.21) |
> | **w fast KM** | **68.68 (↑0.16)** | **65.93 (↑0.18)** | _63.85 (↓1.06)_ | _62.37 (↓0.19)_ | _56.85 (↓1.32)_ | _47.59 (↓0.72)_ |
> | **w Re & KM** | 66.41 (↓2.11) | 63.33 (↓2.42) | 62.43 (↓2.48) | 60.34 (↓2.22) | 55.70 (↓2.47) | 46.05 (↓2.26) |
> | **w Re & fast KM** | _68.52_ | _65.75_ | **64.91** | **62.56** | **58.17** | **48.31** |
>
> [1] Radu Ioan Bot¸ and Dang-Khoa Nguyen. Fast Krasnosel’skii–Mann algorithm with a convergence rate of the fixed point iteration of . SIAM Journal on Numerical Analysis, 61(6):2813–2843, 2023.

---

> ### Author Response · Authors · 2025-11-24
> **3/N: Compared with FedU^2, FedRep and more recent methods (Q2, W1 & W2)**
>
> **1) Compared with FedU^2, FedRep**
>
> **FedU^2** is designed for federated **unsupervised learning**, whereas our work focuses on the supervised setting. **FedRep** targets **personalized federated learning**, aiming to train a personalized model for each client, while our objective is to learn a single shared global model.
>
> Our contribution lies in accelerating the optimization process via relaxation and fast KM techniques, thereby improving the convergence speed. These techniques are in fact **orthogonal** to the goals of FedU^2 and FedRep. One could try to design new methods for unsupervised or personalized FL frameworks based on our acceleration mechanisms to further improve their efficiency; however, such extensions fall outside the scope of this work and constitute promising directions for future research.
>
>
> **2) Compared with more recent methods**
>
> To further demonstrate the advantages of our method, we additionally compare FedRKMGC with several more recent acceleration techniques. Specifically, we include experiments with FedGA24[1], STEM[2], and FedMoS[3] on CIFAR-10 and CIFAR-100 using ResNet-18 under various data distributions. The results are presented in the following table.
> As shown in this table, our method consistently outperforms these recent baselines across all settings, highlighting its robustness and strong acceleration capability.
>
> |        | CIFAR-10               |            |            | CIFAR-100              |            |            |
> |--------|------------------------|------------|------------|------------------------|------------|------------|
> | ACCG%  | IID                    | Dir(0.5)   | Dir(0.3)   | IID                    | Dir(0.5)   | Dir(0.3)   |
> | FedAvg | 89.61(↓3.50)          | 85.95(↓3.90)| 86.33(↓2.64)| 53.17(↓15.35)         | 53.71(↓12.04)| 52.55(↓12.36)|
> | FedProx| 89.50(↓3.61)          | 83.54(↓6.31)| 84.23(↓4.74)| 51.73(↓16.79)         | 51.64(↓14.11)| 51.64(↓13.27)|
> | FedGA  | 89.46(↓3.65)          | 86.51(↓3.34)| 87.29(↓1.68)| 53.46(↓15.06)         | 53.00(↓12.75)| 50.43(↓14.48)|
> | FedGA24| 89.65(↓3.46)          | 83.84(↓6.01)| 83.34(↓5.63)| 53.43(↓15.09)         | 53.04(↓12.71)| 53.67(↓11.24)|
> | STEM   | 89.89(↓3.22)          | 85.74(↓4.11)| 84.96(↓4.01)| 51.90(↓16.62)         | 52.99(↓12.76)| 52.81(↓12.10)|
> | FedMoS | 86.74(↓6.37)          | 86.57(↓3.28)| 88.07(↓0.90)| 50.80(↓17.72)      | 52.13(↓13.62)| 52.11(↓12.80)|
> | SCAFFOLD| 91.93(↓1.18)     | 87.18(↓2.67)| 87.27(↓1.70)| 61.81(↓6.71)          | 61.01(↓4.74)| 60.12(↓4.79)|
> | FedDyn | 91.51(↓1.60)          | 89.26(↓0.59)| 88.01(↓0.96)| 62.73(↓5.79)      | 62.84(↓2.91)| 62.73(↓2.18)|
> | FedADMM| 90.89(↓2.22)          | 86.30(↓3.55)| 87.71(↓1.26)| 63.34(↓5.18)      | 64.29(↓1.46)| 63.18(↓1.73)|
> | **FedRKMGC** | **93.11**        | **89.85**  | **88.97**  | **68.52**             | **65.75**  | **64.91**  |
>
> More discussion can refer to Appendix D.9.
>
> [1] Chenguang Xiao, Zheming Zuo, and Shuo Wang. FedGA: Federated learning with gradient alignment for error asymmetry mitigation, 2024.
>
> [2] Prashant Khanduri, Pranay Sharma, Haibo Yang, Mingyi Hong, Jia Liu, Ketan Rajawat, and Pramod Varshney. STEM: A stochastic two-sided momentum algorithm achieving near-optimal sample and communication complexities for federated learning. In Advances in Neural Information Processing Systems, volume 34, pp. 6050–6061, 2021.
>
> [3] Xiong Wang, Yuxin Chen, Yuqing Li, Xiaofei Liao, Hai Jin, and Bo Li. FedMoS: Taming client drift in federated learning with double momentum and adaptive selection. In IEEE Conference on Computer Communications, pp. 1–10, 2023.

---

> ### Author Response · Authors · 2025-11-24
> **4/N: FedRKMGC under high data heterogeneity (Q2 & W4)**
>
> To further assess the performance stability and convergence advantages of our method, particularly under high data heterogeneity, we additionally conducted comparative experiments on CIFAR-100 with ResNet-18 under Dirichlet partitions of $\alpha=0.1, 0.2$. The results for different methods under different data heterogeneous settings are summarized below.
> These results show that, even in highly heterogeneous settings, our approach maintains a clear advantage. This further confirms the effectiveness and robustness of our method. More details are provided in Appendix D.6.
>
> | ACCG(%)    | IID                | Dir(0.5)             | Dir(0.3)             | Dir(0.2)             | Dir(0.1)               |
> |-----------|---------------------|-----------------------|-----------------------|-----------------------|-------------------------|
> | FedAvg    | 53.17 (↓15.35)     | 53.71 (↓12.04)        | 52.55 (↓12.36)        | 52.38 (↓10.18)        | 48.70 (↓9.47)           |
> | FedProx   | 51.73 (↓16.79)     | 51.64 (↓14.11)        | 51.64 (↓13.27)        | 50.86 (↓11.70)        | 49.33 (↓8.84)           |
> | FedGA     | 53.46 (↓15.06)     | 53.00 (↓12.75)        | 50.43 (↓14.48)        | 53.48 (↓9.08)         | 48.57 (↓9.60)           |
> | SCAFFOLD  | 61.81 (↓6.71)      | 61.01 (↓4.74)         | 60.12 (↓4.79)         | 60.45 (↓2.11)         | _57.80_ (↓0.37)         |
> | FedDyn    | 62.73 (↓5.79)      | _62.84_ (↓2.91)       | 62.73 (↓2.18)         | _61.70_ (↓0.86)       | 57.52 (↓0.65)           |
> | FedADMM   | _63.34_ (↓5.18)    | _64.29_ (↓1.46)       | _63.18_ (↓1.73)       | 61.06 (↓1.50)         | 54.95 (↓3.22)           |
> | **FedRKMGC** | **68.52**       | **65.75**             | **64.91**             | **62.56**             | **58.17**               |

---

> ### Author Response · Authors · 2025-11-24
> **5/N: Cosine similarity for different methods (Q4)**
>
> To provide a more detailed quantitative comparison of different methods, including all ablation variants, we present the per-round average cosine similarity of participating client models on CIFAR-100 with ResNet-18 under Dir(0.1). The results are plotted in Appendix D.5.

---

> ### Author Response · Authors · 2025-11-28
> **Summary**
>
> We sincerely thank the reviewer  for the insightful comments and constructive suggestions. In response, we have added substantial new material to strengthen the paper. Specifically, we have included:
>
> - More experimental details (Q1 & W3) - Appendix D.1
>
> - Additional ablations for FedRKMGC (Q3, Q4, W5 & W6) - Appendix D.6
>
> - Comparisons with more recent methods (Q2, W1 & W2) - Appendix D.9
>
> - Experiments under higher data heterogeneity (Q2 & W4) - Appendix D.7
>
> - Inter-client cosine similarity analysis (Q4) - Appendix D.5
>
> In addition, inspired by the comments from all reviewers, we further added:
>
> - Computation complexity analysis - Appendix B
>
> - Parameter sensitivity analysis - Appendix D.3
>
> - Experiments on non-vision tasks - Appendix D.8
>
> - Motivation for fast KM correction - Appendix A
>
> - Additional study on local epochs for FedADMM - Appendix D.4
>
> Thank you again for your time, effort, and valuable feedback.

---

### Author Response · Authors · 2025-12-04
**Rebuttal Summary**

Dear ACs, SACs and PCs,

Thank you for the time and effort you have dedicated to reviewing our paper. We sincerely appreciate the reviewers’ insightful comments and constructive suggestions. For clarity, we summarize our key responses below.

To address the issues of slow convergence and client drift in FL, inspired by **operator theory (2aiZ), our work presents a **novel, well-motivated (hRr7) and interesting (BYUp) method** by integrating fast KM acceleration and relaxation into federated learning. Our extensive experiments (2aiZ)** further demonstrate that the proposed method achieves **significant communication savings (BYUp)** and superior convergence performance.

In response to the reviewers’ valuable feedback, we have substantially enriched the paper with the following additions:

- More experimental details (hRr7 Q1 & W3; ) - Appendix D.1

- Additional ablations for FedRKMGC (hRr7 Q3, Q4, W5 & W6; ) - Appendix D.6

- Comparisons with more recent methods (hRr7 Q2, W1 & W2; BYUp W2) - Appendix D.9

- Experiments under higher data heterogeneity (hRr7 Q2 & W4; ) - Appendix D.7

- Inter-client cosine similarity analysis (hRr7 Q4; ) - Appendix D.5


- Complexity analysis (2aiZ Q1; BYUp W4) - Appendix B

- Parameter sensitivity analysis (2aiZ Q2 & W2; BYUp W3) - Appendix D.3

- Experiments on non-vision tasks (2aiZ Q3 & W3; BYUp W2) - Appendix D.8

- Further theoretical discussion (2aiZ Q4 & W1; BYUp W1)

- Motivation for fast KM correction (BYUp Q1) - Appendix A

- Study on the choice of local epochs for FedADMM (BYUp Q2) - Appendix D.4

- Comparison with standard KM (BYUp Q3) - Appendix D.6

We hope that these revisions adequately address all concerns and help improve the clarity and quality of the paper.


Best,

Authors

---

### Meta-Review · Area_Chair_iVyd · 2026-01-06

**Summary:**

1. Limited novelty compared to existing works
2. Limited experiments and comparison with other FL methods
3. Lack of theoretical analysis

**Reviewer Concerns:**

Limited novelty and lack of theoretical convergence analysis (pointed by 2 reviewers) not sufficiently addressed.

**Reviewer Scores:**

unchanged

---

### Decision · Program_Chairs · 2026-01-26

Reject